# Understanding Transformers in Time Series Forecasting: A Case Study on MOIRAI

**Dennis Wu**[123]*, **Yihan He**[4]*, **Yuan Cao**[5]*, **Jianqing Fan**[4], **Han Liu**[123]

[1]Northwestern University, [2]Center for Foundation Models and Generative AI,
[3]NSF-Simons AI Institute for the Sky, [4]Princeton University, [5]The University of Hong Kong
```
{hibb, hanliu}@u.northwestern.edu
{yihan.he,jqfan}@princeton.edu
yuancao@hku.hk
```

## Abstract

We give a comprehensive theoretical analysis of transformers as time series prediction models, with a focus on MOIRAI (Woo et al., 2024). We study its approximation and generalization capabilities. First, we demonstrate that there exist transformers that fit an autoregressive model on input univariate time series via gradient descent. We then analyze MOIRAI, one of the state-of-the-art multivariate time series prediction models capable of modeling arbitrary number of covariates. We prove that MOIRAI is capable of automatically fitting autoregressive models with an arbitrary number of covariates, offering insights into its design and empirical success. For generalization, we establish learning bounds for pretraining when the data satisfies Dobrushin's condition. Experiments support our theoretical findings, highlighting the efficacy of using transformers for time series forecasting.

## 1 Introduction

Recent advancement of transformers is reshaping the field of time series forecasting. Numerous studies have demonstrated transformers are an effective architectures for various time series analysis tasks such as forecasting (Woo et al., 2024; Ansari et al., 2024; Liang et al., 2024; Das et al., 2023), anomaly detection (Zhang and Luo, 2025; Wen et al., 2025) and more. While recent works have made efforts to enhance transformers for times series forecasting by modifying model architecture (Zhang and Yan, 2023; Liu et al., 2023; Wu et al., 2021) or data processing (Reneau et al., 2023; Nie et al., 2022), the research community has yet to understand how transformers perform so well on forecasting tasks even on the simplest settings. Moreover, existing methods (Zhang and Yan, 2023; Wu et al., 2021; Woo et al., 2024; Liu et al., 2023) predominantly depend on heuristic reasoning and are notably deficient in rigorous theoretical analysis. A deeper understanding of transformers in time series forecasting will guide the design of more effective architectures by future practitioners.

A critical challenge for transformers to handle time series data is to handle arbitrary number of covariates as the architecture proposed in Vaswani et al. (2017) is designed to handle a fixed vocabulary size. Therefore, recent studies have developed several ways to address this issue. MOIRAI (Woo et al., 2024) propose an unified way to concatenate all covariates into a single long univariate time series (any-variate encoding). MOIRAI also propose a novel any-variate mechanism for disambiguating different covariates. iTransformers (Liu et al., 2023) treat each covariate as tokens. Compared to traditional approaches that perform attention along the temporal dimension, they perform attention along the variate attention Other models choose to discard covariate features by only considering univariate time series (Ansari et al., 2024; Rasul et al., 2023). In this paper, we start by theoretically analyzing MOIRAI and its novel mechanisms. The main goal of this paper is to address the central question: *What characteristics of architectures in MOIRAI contribute to their strong performance on time series data?*

To understand MOIRAI in time series learning, we begin by examining one of the most widely used algorithms for time series regression: the auto-regressive (AR) regression (Hamilton, 2020). **1.** We show that transformers are indeed capable of performing AR regression on univariate time series, indicating they can handle time series in a principled, algorithmic fashion. **2.** We apply our theoretical

---

*equal contribution

results on a state-of-the-art time series model, MOIRAI (Woo et al., 2024), a transformer-based model that can handle arbitrary number of covariates in various time series tasks. While several novel designs in MOIRAI are for engineering purpose and heuristic, we show that these unique designs enable it to perform AR regression on arbitrary number of covariates. **3.** Our theoretical results contribute to understand the effectiveness of recent time series pretraining approaches (Ansari et al., 2024; Woo et al., 2024; Jin et al., 2023; Das et al., 2023; Rasul et al., 2023; Liang et al., 2024). These methods involve pretraining transformers on large collections of time series, often spanning diverse domains, and have demonstrated strong empirical performance on a wide range of domains. To analyze this phenomenon, we derive generalization error bounds for pretrained transformer models under a formal statistical framework. The contributions of this paper is threefold:

- From an algorithmic approximation view, we prove the existence of a transformer capable of fitting an AR model on any given univariate time series via gradient descent. When extending this to the multi-variate setting, we theoretically verify that MOIRAI, with its novel any-variate attention can automatically adjust the dimensionality of the AR model to fit time series with an arbitrary number of covariates. Our approximation results not only explain the strong performance of MOIRAI, but also justify the encoding and architecture design of MOIRAI.
- We present the first pretraining generalization bound for MOIRAI on time series learning. We show that when the pretraining data satisfies Dobrushin's condition, the test error can be effectively bounded even when the data does not satisfy i.i.d. Specifically, when pretraining MOIRAI on $n$ multi-variate time series with length $T$, the test error decays by a rate of $1/\sqrt{nT}$.
- Our experimental results on both synthetic and real-world data match our theories by showing that the prediction error of transformers reduces as the input time series length increases, corresponding to our approximation result. This empirical finding not only verifies our theoretical results, but also demonstrates transformers are capable of acquiring algorithmic capability through training.

**Organization.** Section 2 introduces our problem setup and preliminaries. Section 3 describes how transformers simulate different AR regression algorithms and a case study on MOIRAI. Section 4 analyzes the pretraining guarantee for learning transformers over different time series. Section 5 conducts experiments to verify our theoretical results on synthetic and real-world datasets. Section 6 discusses how our findings generalize to other models and limitations. Additional Related works are discussed in the Appendix C.

**Notations.** We use the following notation conventions. The vector-valued variable is given by boldfaced characters. We denote $[n] := \{1, \dots, n\}$ and $[i : j] := \{i, i + 1, \dots, j\}$ for $i < j$. The universal constants are given by $C$ and are ad hoc. Considering a sequence of vectors $(\boldsymbol{x}_1, \cdots, \boldsymbol{x}_T)$, we use $\boldsymbol{x}$ without index to represent the whole sequence, and $\boldsymbol{x}_{i:j}$ represents $(\boldsymbol{x}_i, \cdots, \boldsymbol{x}_j)$ for $i < j$. We impose periodic boundary conditions for the negative index, i.e., $\boldsymbol{x}_{-1} = \boldsymbol{x}_T$. For a vector $\boldsymbol{v}$ we denote $\|\boldsymbol{v}\|_2$ as its $L_2$ norm. For a matrix $\boldsymbol{A} \in \mathbb{R}^{m \times n}$ we denote its operator norm as $\|\boldsymbol{A}\|_2 := \sup_{\boldsymbol{v} \in \mathbb{S}^{n-1}} \|\boldsymbol{A}\boldsymbol{v}\|_2$. Random variables are given by calligraphic characters $\mathcal{X}$, and elements from a domain set are given by normal font x. For more details, see Table 1.

## 2 PROBLEM SETUP

This section presents backgrounds and formula definitions of the transformer model, and then introduce the auto-regressive model.

**Transformers.** We consider a sequence of $N$ input vectors $\{h_i\}_{i=1}^N \subset \mathbb{R}^D$. We introduce $\boldsymbol{H} := [h_1, \cdots, h_N] \in \mathbb{R}^{D \times N}$. Given any $\boldsymbol{H} \in \mathbb{R}^{D \times N}$, we define the attention layer as follows.

**Definition 2.1** (Attention layer). *A self-attention layer with $M$ heads is denoted as $\text{Attn}^{\dagger}_{\boldsymbol{\theta}_0}(\cdot)$ with parameters $\boldsymbol{\theta}_0 = \{(\boldsymbol{V}_m), (\boldsymbol{Q}_m), (\boldsymbol{K}_m)\}_{m \in [M]} \subset \mathbb{R}^{D \times D}$. The self-attention layer processes any given input sequence $\boldsymbol{H} \in \mathbb{R}^{D \times N}$ as*

$$Attn^{\dagger}_{\boldsymbol{\theta}_0}(\boldsymbol{H}) := \boldsymbol{H} + \frac{1}{N} \sum_{m=1}^M (\boldsymbol{V}_m \boldsymbol{H}) \times \sigma\left((\boldsymbol{Q}_m \boldsymbol{H})^{\top}(\boldsymbol{K}_m \boldsymbol{H})\right),$$

*where $\sigma(t) := ReLU(t)/N$ is the ReLU function normalized by $N$.*

Next, we introduce the any-variate attention, where Woo et al. (2024) uses it to replace the standard attention in transformers. The any-variate attention introduces two learnable variables: Attention

Bias $u_1, u_2 \in \mathbb{R}$, for disambiguation between variates. In (Woo et al., 2024), they use this mechanism to allow transformers to distinguish different covariates.

**Definition 2.2** (Any-variate Attention.). *An any-variate attention layer with $M$ heads is denoted as $Attn_{\boldsymbol{\theta}_1}(\cdot)$ with parameters $\boldsymbol{\theta}_1 = \{(\boldsymbol{V}_m), (\boldsymbol{Q}_m), (\boldsymbol{K}_m), (u_m^1), (u_m^2)\}_{m \in [M]}$. With any input $H \in \mathbb{R}^{D \times N}$, we have*

$$Attn_{\boldsymbol{\theta}_1}(\boldsymbol{H}) := \boldsymbol{H} + \frac{1}{N} \sum_{m=1}^{M} (\boldsymbol{V}_m \boldsymbol{H}) \times \sigma \left( (\boldsymbol{Q}_m \boldsymbol{H})^\top (\boldsymbol{K}_m \boldsymbol{H}) + u_m^1 * \boldsymbol{U} + u_m^2 * \bar{\boldsymbol{U}} \right),$$

*where $\sigma(t) = ReLU(t)/N$. $\boldsymbol{U} \in \mathbb{R}^{N \times N}$ is a block diagonal matrix with block size $T \in \mathbb{N}^+$, such that each block consists of $1$s, $\bar{\boldsymbol{U}} = \boldsymbol{I} - \boldsymbol{U}$, and $*$ denotes element-wise multiplication.*

**Definition 2.3** (MLP Layer). We denote an MLP layer with hidden state dimension $D'$ as $\text{MLP}_{\boldsymbol{\theta}}(\cdot)$ with parameters $\boldsymbol{\theta}_2 = (\boldsymbol{W}_1, \boldsymbol{W}_2) \in \mathbb{R}^{D' \times D} \times \mathbb{R}^{D \times D'}$. The MLP layer processes any given input sequence $\boldsymbol{H} \in \mathbb{R}^{D \times N}$ as $\text{MLP}_{\boldsymbol{\theta}_2}(\boldsymbol{H}) := \boldsymbol{H} + \boldsymbol{W}_2 \sigma(\boldsymbol{W}_1 \boldsymbol{H})$.

Finally, we define a $L$-layer transformer.

**Definition 2.4** (MOIRAI Transformer). *We define the $L$-layer MOIRAI transformer (Woo et al., 2024), $TF_{\boldsymbol{\theta}}(\cdot)$, as*

$$TF_{\boldsymbol{\theta}}(\boldsymbol{H}) = MLP_{\boldsymbol{\theta}_2^L} \left( Attn_{\boldsymbol{\theta}_1^L} \left( \cdots MLP_{\boldsymbol{\theta}_2^1} \left( Attn_{\boldsymbol{\theta}_1^1}(\boldsymbol{H}) \right) \right) \right).$$

*Note that this transformer is equipped with any-variate attention instead of the standard attention. For transformers with standard attention, we denote it as $TF_{\boldsymbol{\theta}}^\dagger(\cdot)$.*

We use $\boldsymbol{\theta}$ to denote the vectorization of all parameters in a transformer and super-index $\ell$ to denote the parameter of the $\ell$-th layer. Thus, the parameter of a transformer is defined by

$$\boldsymbol{\theta} = \left\{ \left\{ \left( \{\boldsymbol{Q}_m^\ell, \boldsymbol{K}_m^\ell, \boldsymbol{V}_m^\ell, u_m^{1,\ell}, u_m^{2,\ell}\}_{m \in [M]}, \boldsymbol{W}_1^\ell, \boldsymbol{W}_2^\ell \right) \right\}_{\ell \in [L]} \right\}.$$

We denote the "attention-only" transformers with $\boldsymbol{W}_1^{(\ell)}, \boldsymbol{W}_2^{(\ell)} = 0$, as $TF_{\boldsymbol{\theta}}^0(\cdot)$ for shorthand. We define the following norm of a MOIRAI transformer as

$$\|\boldsymbol{\theta}\|_{op} := \max_{\ell \in [L]} \left\{ \max_{m \in [M^\ell]} \left\{ \|\boldsymbol{Q}_m^\ell\|_2, \|\boldsymbol{K}_m^\ell\|_2, |u_m^{1\ell}|, |u_m^{2\ell}| \right\} + \sum_{m=1}^{M^\ell} \|\boldsymbol{V}_m^\ell\|_2 + \|\boldsymbol{W}_1^\ell\|_2 + \|\boldsymbol{W}_2^\ell\|_2 \right\},$$

where $M^\ell$ is the number of heads of the $\ell$-th Attention layer.

**Auto-regressive Model.** Here, we first consider the case where we aim to find a multi-layered transformer that performs regression via In-context learning (ICL). Specifically, we assume our data is generated from an autoregressive process $\text{AR}_d(q)$ as follows, where $q, d$ denotes the steps of lag and number of covariates, respectively. Consider a sequence of data $\boldsymbol{x} \in \mathbb{R}^{d \times T} := (\boldsymbol{x}_1, \ldots, \boldsymbol{x}_T)$, where $\boldsymbol{x}_t = (x_t^1, \cdots, x_t^d) \in \mathbb{R}^d$. Assuming our target (variate of interest) is univariate, we define the $\text{AR}_d(q)$ process generates $x_t^1$ as follows:

$$x_t^1 = \sum_{i=1}^{q} \sum_{j=1}^{d} a_i^j \cdot x_{t-i}^j + \epsilon_t = \sum_{j=1}^{d} \langle \boldsymbol{w}^j, \boldsymbol{x}_{t-q:t-1}^j \rangle + \epsilon_t, \tag{2.1}$$

where $\epsilon_t \sim N(0, 1)$, $a_i^j \in \mathbb{R}^1$. We denote the concatenation of all weights $\boldsymbol{w}^\star = (\boldsymbol{w}_1, \cdots, \boldsymbol{w}_d) \in \mathbb{R}^{qd}$. We assume bounded features $\|\boldsymbol{x}_{t-q:t-1}\|_2 \leq B_x$, for all $t \in [T]$. The first equation writes the $\text{AR}$ process in scalar form, and the second writes it in vector form. In the following chapters, we will start by considering the uni-variate case ($\text{AR}_1$) and then move on to the multi-variate case ($\text{AR}_d$).

## 3 Transformers Simulate Auto-Regressive Regression

We first investigate how MOIRAI performs univariate AR regression. Next, we will move on to multi-variate AR regression and analyze how MOIRAI's unique design and pre-processing methods enable its advantages.

## 3.1 Warm Up: Univariate Autoregressive Regression

We start our analysis with a warm-up example on the $\mathtt{AR}_1(q)$ model. We show that standard transformers are capable of performing gradient descent via in-context learning on autoregressive data. The results in this subsection apply to univariate time series models such as (Ansari et al., 2024). Here, we consider an input sequence with the following form

$$\boldsymbol{H} := \begin{bmatrix} x_1 & x_2 & \ldots & x_T & 0 \\ \boldsymbol{p}_1 & \boldsymbol{p}_2 & \ldots & \boldsymbol{p}_T & \boldsymbol{p}_{T+1} \end{bmatrix} \in \mathbb{R}^{D \times (T+1)}, \quad \boldsymbol{p}_i := \begin{bmatrix} \boldsymbol{0}_{d'} \\ \boldsymbol{e}_i \\ 1 \\ 1\{i < T\} \end{bmatrix} \in \mathbb{R}^{d'+T+3}, \quad (3.1)$$

where $\boldsymbol{e}_i$ is an one-hot vector with 1 at the $i$-th entry, and $d' + T + 3 = D$. Here, our goal is to predict $x_{T+1}$.

**Remark 3.1.** *Most in-context learning studies (Akyürek et al., 2023; Bai et al., 2024; Li et al., 2023) make an assumption on the input data, where they assume it is formatted with features and labels in the same column, i.e.,* $\begin{bmatrix} \boldsymbol{x}_1 & \boldsymbol{x}_2 & \ldots & \boldsymbol{x}_N \\ \boldsymbol{y}_1 & \boldsymbol{y}_2 & \ldots & \boldsymbol{y}_N \end{bmatrix}$. *In contrast, we adopt a natural approach that leverages the raw structure of the data, particularly for the* $\mathtt{AR}_d(q)$ *process. In this setting, each time step's label also serves as a feature for future steps. Further, the unknown value of* $q$ *complicates the task of achieving such a format in Remark 3.1.*

Our next lemma shows that transformers are indeed capable of reformatting $\boldsymbol{H}$ into the form in Remark 3.1. Notably, we relax the assumption in Remark 3.1 of previous studies as well.

**Lemma 3.2.** *Given a sequence of token $\boldsymbol{H}$ in the form of Equation 3.1, there exists a one-layer, $q_{\max}$ head attention layer, such that for any $q \leq q_{\max}$, the columns of $Attn_{\boldsymbol{\theta}}^{\dagger}(\boldsymbol{H})$ has the following form:*

$$Attn_{\boldsymbol{\theta}_1}(\boldsymbol{H})_i := \begin{bmatrix} x_i & x_{i-1} & \cdots & x_{i-q} & \boldsymbol{p}_i' \end{bmatrix}^{\top}, \quad \boldsymbol{p}_i' := \begin{bmatrix} \boldsymbol{0}_{d'-q} \\ \boldsymbol{e}_i \\ 1 \\ 1\{i < T\} \end{bmatrix}. \quad (3.2)$$

The proof is in Appendix E.1. Lemma 3.2 is crucial in our analysis as it connects the theoretical results in ICL (Bai et al., 2024) to univariate time series forecasting. When data formats in the form of Remark 3.1, Bai et al. (2024) show that there exists a multi-layer transformer that performs linear regression via gradient descent on the first $N - 1$ data points and evaluates the $N$-th one. Thus, Lemma 3.2 implies transformers are also capable of performing linear regression on time series data.

This lemma applies to both any-variate attention and standard attention, as the latter can be viewed as a special case of any-variate attention by setting $u^1, u^2 = 0$. Additionally, the construction of a single layer with $q$ heads is not a strict requirement; the lemma also holds for $c$ layers of $q/c$ head attention, for any $c$ satisfies $q/c \geq 2$.

With Lemma 3.2, we are able to apply the in-context learning results in (Bai et al., 2024) on the $\mathtt{AR}_1(q)$ case. Consider the data generated by the $\mathtt{AR}$ process in Equation 2.1. Given an input time series $\boldsymbol{x} \in \mathbb{R}^{d \times T}$, we define the least squares estimator as the empirical risk minimizer over the time series, i.e.,

$$\ell_{\mathrm{reg}}(\boldsymbol{w}, \boldsymbol{x}_{t-1:t-q}) := \frac{\left[ \langle \boldsymbol{w}, [\boldsymbol{x}_{t-1:t-q}^1; \ldots; \boldsymbol{x}_{t-1:t-q}^d] \rangle - x_t^1 \right]^2}{2}$$

$$L_{\mathrm{reg}}(\boldsymbol{w}, \boldsymbol{x}) := \frac{1}{T-1} \sum_{t=1}^{T-1} \ell_{\mathrm{reg}}(\boldsymbol{w}, \boldsymbol{x}_{t-1:t-q}), \quad \widehat{\boldsymbol{w}}_{\mathrm{ERM}} := \underset{\boldsymbol{w} \in \mathbb{R}^{dq}}{\operatorname{argmin}} \, L_{\mathrm{reg}}(\boldsymbol{w}, \boldsymbol{x}),$$

where $[\boldsymbol{v}; \boldsymbol{u}]$ denotes the concatenation between vectors, as $[\boldsymbol{x}_{t-1:t-q}^1; \boldsymbol{x}_{t-1:t-q}^2] = (\boldsymbol{x}_{t-1}^1, \boldsymbol{x}_{t-2}^1, \cdots, \boldsymbol{x}_{t-q+1}^2, \boldsymbol{x}_{t-q}^2) \in \mathbb{R}^{2q}$, $\widetilde{\boldsymbol{x}}$ denotes masking out the last time step of the target variate, and $L_{reg}$ is a loss, which is $\alpha$-strongly convex, and $\beta$-smooth over $\mathbb{R}^{dq}$. We make the following assumption and then present our first result on univariate time series ($d = 1$).

**Proposition 3.3** (Univariate Autoregressive Regression via MOIRAI)**.** *Assume the regression problem is well-defined and has a bounded solution and fix a $q_{\max} > 0$. For any $0 \leq \alpha \leq \beta$ with $\kappa := \frac{\beta}{\alpha}$,*

$B_w > 0$, and $\epsilon < B_x B_w / 2$, there exists a $L$-layer MOIRAI transformer $TF_{\boldsymbol{\theta}}^0(\cdot)$, with

$$L = L_1 + L_2, \quad L_1 = \lceil 2\kappa \log(\frac{B_x B_w}{2\epsilon}) \rceil, \ L_2 = \lceil \frac{q_{\max}}{3} \rceil, \ _{max}\ell \in [L] M^{(\ell)} \le 3, \ ||\boldsymbol{\theta}||_{op} \le |4R + 8\beta^{-1}|,$$

($R := max\{B_x B_w, B_x, 1\}$), the following holds. On any input data $\boldsymbol{x}$ generated by any $\mathtt{AR}_1(q)$ process such that $0 < q \le q_{\max}$, $||\widehat{\boldsymbol{w}}_{ERM}||_2 \le {}^{B_w}/2$, we have

$$\|\widehat{\boldsymbol{x}}_T - \left\langle \widehat{\boldsymbol{w}}_{ERM}, [\boldsymbol{x}_{t-1:t-q}^1; \ldots; \boldsymbol{x}_{t-1:t-q}^d] \right\rangle\| \le \epsilon, \tag{3.3}$$

where $\widehat{\boldsymbol{x}}_T = \mathtt{read}(TF_{\boldsymbol{\theta}}^0(\boldsymbol{H}))$. $\mathtt{read}(\boldsymbol{H})$ operation reads out the first entry of $T$-th column of $\boldsymbol{H}$.

This proposition follows immediately from Lemma 3.2 and (Bai et al., 2024, Theorem 4). The above result applies for standard transformers as well as in our construction $u_m^1, u_m^2 = 0$ in all heads and layers. Further, one can replace the least squares ERM with lasso or ridge ERM and obtain a similar result by applying Theorem 4, 7, and 13 of (Bai et al., 2024).

So far, we show that transformers are capable of solving univariate AR regression with, at best, one additional layer compared to the results in (Bai et al., 2024). The result above provides insights on transformer-based univariate time series foundation models (Ansari et al., 2024; Rasul et al., 2023; Das et al., 2023). To study MOIRAI, we then include two ingredients into our analysis: the *any-variate encoding* and the *covariates* in the following chapters.

## 3.2 CASE STUDY: MOIRAI TRANSFORMER

In this subsection, we extend our results to the multivariate autoregressive process ($d > 1$) and the encoding method of MOIRAI. Note that in the multi-variate case, we focus on MOIRAI as it is compatible with arbitrary numbers of covariates. For other transformer-based models with the same feature, we leave it to future work. We start by introducing the any-variate encoding.

**Any-Variate Encoding.** Woo et al. (2024) propose to flatten a $d$-dimensional time series, $\boldsymbol{x} \in \mathbb{R}^{d \times T}$, into a 1-dimensional sequence, i.e., $\boldsymbol{x}' \in \mathbb{R}^{1 \times Td}$. This operation transforms time series with arbitrary number of covariates ($d$), into a long sequence with fixed dimension, enabling consistent input dimension for transformers. Following the flattening operation, Woo et al. (2024) also proposes to add two types of indices into the input sequence: the time and variate ID. We term the above operations as the any-variate encoding, which transforms a multivariate sequence $\boldsymbol{x} \in \mathbb{R}^{d \times T}$, as follows:

$$\begin{bmatrix} x_1^1 & \cdots & x_T^1 \\ x_1^2 & \cdots & x_T^2 \\ \vdots & \vdots & \vdots \\ x_1^d & \cdots & x_T^d \end{bmatrix} \rightarrow \begin{bmatrix} x_1^1 & \cdots & x_T^1 & \cdots & x_1^d & \cdots & x_T^d \\ \boldsymbol{p}_1 & \cdots & \boldsymbol{p}_T & \cdots & \boldsymbol{p}_1 & \cdots & \boldsymbol{p}_T \\ \boldsymbol{e}_1 & \cdots & \boldsymbol{e}_1 & \cdots & \boldsymbol{e}_d & \cdots & \boldsymbol{e}_d \end{bmatrix}, \tag{3.4}$$

where $\boldsymbol{e}_i$ is the variate index, a one-hot vector with $i$-th entry being 1, and $\boldsymbol{p}_i$ is the time index, which is defined the same as Equation (3.1). This is without loss of generality because the discrete-time and variate ID used in (Woo et al., 2024) can be easily transformed into our constructed positional encoding with the patch embedding layer. Note that only the target variate has length $T$, we highlight $x_T^1$ as it is our prediction target and will be masked as 0.

Now we define the history matrix $\mathtt{A}_i(q) \in \mathbb{R}^{q+1 \times T}$ for the $i$-th covariates $(x_1^i, \cdots, x_T^i)$, with order $q$, such that

$$\mathtt{A}_i(q)_{\mu,\nu} := x_{\nu-\mu+1}^i, \quad \text{for } \mu \in [d], \nu \in [q],$$

where in the $j$-th column of $\mathtt{A}_i(q)$, it contains historical values of $x_j^i$ with lag $q > 0$.

**Lemma 3.4.** *Fix $q_{\max}, D \in \mathbb{N}^+$. Given any $T > 0, d' > q > 0, d > 0$ such that $T > q$, $q_{\max} \ge q$. For any input matrix $\boldsymbol{H}$ in the form of any-variate encoding in Equation 3.4, such that $\boldsymbol{H} \in \mathbb{R}^{D \times dT}$. There exists a one layer, $q_{\max}$ head **any-variate attention** that performs the following operation.*

$$\begin{bmatrix} x_1^1 & \cdots & x_T^1 & x_1^2 & \cdots & x_T^2 & \cdots & x_1^d & \cdots & x_T^d \\ \boldsymbol{p}_1 & \cdots & \boldsymbol{p}_T & \boldsymbol{p}_1 & \cdots & \boldsymbol{p}_T & \cdots & \boldsymbol{p}_1 & \cdots & \boldsymbol{p}_T \\ \boldsymbol{e}_1 & \cdots & \boldsymbol{e}_1 & \boldsymbol{e}_2 & \cdots & \boldsymbol{e}_2 & \cdots & \boldsymbol{e}_d & \cdots & \boldsymbol{e}_d \end{bmatrix} \mapsto \begin{bmatrix} \mathtt{A}_1(q) & \mathtt{A}_2(q) & \cdots & \mathtt{A}_d(q) \\ \boldsymbol{0}_{d'' \times T} & \boldsymbol{0}_{d'' \times T} & \cdots & \boldsymbol{0}_{d'' \times T} \\ \ddots & \ddots & \cdots & \ddots \end{bmatrix},$$

*where $d'' = d' - q_{\max}$.*

The proof is in Appendix E.2. Intuitively, the above operation performs the same operation in Lemma 3.2 but in a variate-wise fashion. Lemma 3.4 is crucial to our understanding to MOIRAI as it shows that any-variate attention is capable of organizing the history of each variate efficiently and in a parallel way. To again achieve the format in Remark 3.1, one has to stack all $\texttt{A}_i(q)$ in the same columns, which can be easily done by a single layer of attention via Lemma 3.2 and (Bai et al., 2024, Proposition A.5) (details in Appendix E.2). This lemma serves as a foundation for MOIRAI to handle multi-variate time series with in-context learning which we present as the theorem below.

**Remark 3.5.** *Comparing to Lemma 3.2, Lemma 3.4 is specifically for any-variate attention in our construction, where we demonstrate that several special mechanisms in any-variate attention enables variate-wise operations in parallel.*

**Theorem 3.6** (Any-variate Autoregressive Regression via MOIRAI). *Let $L_1 = \lceil \frac{q_{\max}}{3} \rceil + 1$, $L_2 = \lceil 2\kappa \log \frac{B_x B_w}{2\epsilon} \rceil$, with some $d_{\max}, q_{\max} > 0$. For any $0 \leq \alpha \leq \beta$ with $\kappa := \frac{\beta}{\alpha}$, $B_w > 0$, and $\epsilon < B_x B_w / 2$, there exists an $(L_1 + L_2)$-layer of MOIRAI transformer equipped with any-variate Attention, satisfies the following*

$$\max_{\ell \in [L_1 + 1, L_2]} M^{(\ell)} \leq 3, \ \|\|\boldsymbol{\theta}\|\| \leq |4R + 8\beta^{-1}|, \quad \sum_{\ell=1}^{L_1} M^{(\ell)} = d_{\max} + q_{\max}, \quad D = (q_{\max} + 1)d_{\max} + T + 2,$$

*such that for any input time series $\boldsymbol{x}$ with length $T$ generated from an $\texttt{AR}_d(q)$ process, where $\boldsymbol{x} \in \mathbb{R}^{d \times T}$, $q \leq q_{\max}$, $d \leq d_{\max}$, it satisfies*

$$\|\widehat{\boldsymbol{x}}_T^1 - \left\langle \boldsymbol{w}_i^\star, [\boldsymbol{x}_{T-1:T-q}^1; \ldots; \boldsymbol{x}_{T-1:T-q}^d] \right\rangle\| \leq \epsilon, \tag{3.5}$$

*where $\widehat{\boldsymbol{x}}_T^1 = \texttt{read}(TF_{\boldsymbol{\theta}}^0(\boldsymbol{H}))$, and $\boldsymbol{H} \in \mathbb{R}^{D \times N}$ is the any-variate encoding of $\boldsymbol{x}$.*

**Remark 3.7.** *Theorem 3.6 indicates there exists a MOIRAI transformer that fits an autoregressive model on time series as long as the number of covariates no greater than $d_{\max}$ and lags no greater than $q_{\max}$. This shows its ability to infer the underlying AR model in a principled way and provides a possible explanation for its zero-shot performance on a wide range of datasets. To modify the setting from multi-variate prediction, one can easily apply Bai et al. (2024, proposition A.5), where the hidden dimension scales linearly w.r.t. the number of variates to predict.*

**Remark 3.8.** *Here the max number of lags and covariates depends on the hidden size $D$. If MOIRAI takes more than $d_{\max}$ covariates, in our construction, it will only fit AR with the first $d_{\max}$ covariates. Therefore, while exceeding $d_{\max}$ will not completely destroy model performance, a certain performance degradation is expected. The same concept applies to $q_{\max}$ as well. However, we would like to highlight a counterexample that one can find an AR model that has non-zero weights on $> q_{\max}$ lags, in this case, the model might not be able to generate meaningful prediction.*

The proof is in Appendix D. Observe that there exist two trade-offs in Theorem 3.6. First, $q_{\max}d_{\max}$ is upper bounded by the $D$ (up to constant), which is a natural trade-off in our construction. Second, the approximation error is roughly $O(e^{-L})$, suppressed exponentially by the number of layers, as in our analysis, each layer of MOIRAI performs a single step of gradient descent on $L_{\text{reg}}$.

## 4 ANALYSIS ON PRETRAINING

In this section, we investigate the generalization bound of pretraining MOIRAI on multiple non IID time series. Specifically, we assume the time series satisfy the Dobrushin's condition, which is a regularity condition commonly used for dependent data. Informally, we show that the generalization error decays roughly with the order of $\mathcal{O}(\frac{1}{\sqrt{nT}})$, where $n$ is the number of time series and $T$ is the length of each time series. This provides insights and justifications on the large scale pretraining for time series transformer-based models.

Let $\pi$ be a meta distribution, and each distribution drawn from it $\texttt{P}^{(T)} \sim \pi$, satisfies Dobrushin's condition (Dobrushin, 1968) (which we will introduce shortly). For pretraining data, we first sample $n$ distributions $\texttt{P}_j^{(T)}$ i.i.d. from $\pi$, and for each distribution, we sample a time series $(\boldsymbol{x}_{1j}, \cdots, \boldsymbol{x}_{Tj})$, for $j \in [n]$, and each of them contains no more than $d$ covariates and with lag step no more than $q$.

For each time series, we encode it with any-variate encoding into an input matrix denoted as $\boldsymbol{H} \in \mathbb{R}^{D \times N}$.[1] We define each pretraining sample as $\boldsymbol{z}_j := (\boldsymbol{H}_j, y_j)$, where $y_j = \boldsymbol{x}_{Tj}^1$. We consider

---

[1] Due to any-variate encoding, $N = dT$.

the squared loss between model prediction and the label, i.e.

$$\ell(\boldsymbol{z}_t, \boldsymbol{\theta}) := \frac{1}{2}\Big[y_t - \mathtt{Clip}_{B_x}\Big(\mathtt{read}_y\Big(\mathrm{TF}_{\boldsymbol{\theta}}^R(\boldsymbol{H})\Big)\Big)\Big]^2,$$

where $\mathtt{Clip}_{B_x}(t) := \max\{\min\{t, B_x\}, -B_x\}$, and $\mathrm{TF}_{\boldsymbol{\theta}}^R$ is the MOIRAI transformer defined in Definition 2.4 with $\mathtt{Clip}(\cdot)$ applied after each layer. The pretraining loss and test loss is defined as the following:

$$\widehat{L}(\boldsymbol{\theta}) := \frac{1}{nT}\sum_{t=1}^T \sum_{j=1}^n \ell(\boldsymbol{\theta}, \boldsymbol{z}_{jt}), \ L(\boldsymbol{\theta}) := \mathbb{E}_{\boldsymbol{z}, \mathrm{P}^{(T)}}\left[\ell(\boldsymbol{\theta}, \boldsymbol{z})\right]. \tag{4.1}$$

The goal of our pretraining algorithm is to find an empirical risk minimizer (ERM) over MOIRAI transformers with $L$ layers, $M$ heads, and norm bounded by $B$:

$$\widehat{\boldsymbol{\theta}} := \underset{\boldsymbol{\theta} \in \Theta_{L,M,D',B}}{\operatorname{argmin}} \widehat{L}(\boldsymbol{\theta}),$$

$$\Theta_{L,M,D',B} := \Big\{\boldsymbol{\theta} = \Big(\boldsymbol{\theta}_1^{(1:L)}, \boldsymbol{\theta}_2^{(1:L)}\Big) : \max_{\ell \in [L]} M^{(\ell)} \le M, \ \max_{\ell \in [L]} D^{(\ell)} \le D', \ \|\boldsymbol{\theta}\|_{op} \le B\Big\}. \tag{4.2}$$

### 4.1 NON IID TIME SERIES

Here, we consider the training data $\boldsymbol{x}$ to be drawn from a distribution $\mathrm{P}$ satisfying Dobrushin's condition. Under this condition, we are able to present several generalization bounds on pretraining.

**Definition 4.1** (Dobrushin's Uniqueness Condition). *Let $\mathcal{X} = (\mathcal{X}_1, \cdots, \mathcal{X}_T)$ be a sequence of random variables over $\mathcal{D}_{\mathcal{X}}^T$. The Dobrushin coefficient $\alpha$ of $\mathcal{X}$ and the influence $\boldsymbol{I}$ of variable $\mathcal{X}_j$ on variable $\mathcal{X}_i$ are defined as*

$$\alpha(\mathcal{X}) := \max_{1 \le i \le T} \sum_{j \ne i} \boldsymbol{I}_{j \to i}(\mathcal{X}), \quad \boldsymbol{I}_{j \to i}(\mathcal{X}) := \max_{x_{-i-j}, x_j, x_j'} \big|\big|P_{\mathcal{X}_i|\mathcal{X}_{-i}}\left(\cdot|x_{-i-j}, x_j\right), P_{\mathcal{X}_i|\mathcal{X}_{-i}}\left(\cdot|x_{-i-j}, x_j'\right)\big|\big|_{TV},$$

*where $x_{-i-j} \in \mathcal{D}_{\mathcal{X}}^{T-2}, x_j, x_j' \in \mathcal{D}_{\mathcal{X}}, ||\cdot||_{TV}$ denotes the total variation distance, and $x_{-i}$ represents the vector $\boldsymbol{x}$ after omitting the $i$-th element. We say the variable satisfies Dobrushin's uniqueness condition if $\alpha(\mathcal{X}) < 1$. For a distribution $\mathrm{P}$, we denote $\alpha(\mathrm{P}) = \sup_{\mathcal{X} \sim \mathrm{P}} \alpha(\mathcal{X})$.*

**Definition 4.2** (Log Dobrushin's Coefficients). *Let $\mathcal{X} = (\mathcal{X}_1, \cdots, \mathcal{X}_T)$ be a random variable over $\mathcal{D}_{\mathcal{X}}^T$ and let $\mathrm{P}_z$ denote its density. Assume that $\mathrm{P}_z > 0$ on all $\Omega^T$. For any $i \ne j \in [T]$, the log influence between $j$ and $i$ is defined as:*

$$I_{j,i}^{\log}(\mathcal{X}) = \frac{1}{4} \sup \log \frac{P\left[x_i, x_j, x_{-i-j}\right] P\left[x_i', x_j', x_{-i-j}\right]}{P\left[x_i', x_j, x_{-i-j}\right] P\left[x_i, x_j', x_{-i-j}\right]},$$

*where the $\sup$ is taken over $x_{-i-j}, x_i, x_i', x_j, x_j'$, and the log-coefficient of $\mathcal{X}$ is defined as $\alpha_{\log}(\mathcal{X}) = \max_{i \in [T]} \sum_{j \ne i} I_{j,i}^{\log}(\mathcal{X})$. Note that $\alpha(\cdot)$ has a natural bound in $[0, T-1]$, with $\alpha = 0$ implies i.i.d.*

**Remark 4.3** (On the choice of Dobrushin's condition). *In general, time series analysis (non IID) requires certain level of regularity condition on data, to derive meaningful learning bounds and exclude extream cases. For example, mixing properties (Mohri and Rostamizadeh, 2010; Steinwart and Christmann, 2009; Kuznetsov and Mohri, 2014), hypercontractivity (Ziemann and Tu, 2022), or Dobrushin's condition (Dagan et al., 2019). Dobrushin's condition is not only a commonly used regularity condition for time series data, it also fits several common statistical models, which makes our derivation suitable for further analysis on different models. An example is shown in our section 4.3. One more benefit of having this condition instead of mixing properties is that we do not need to assume stationarity of the data, which can easily be violated in practice. We highlight that verifying all those conditions is difficult in practice, as the true distribution of data is intractable.*

### 4.2 GENERALIZATION BOUNDS OF MOIRAI

**Theorem 4.4** (Pretraining Generalization Bound). *Let $\Theta_{L,M,D',B}$ be the parameter space defined in Equation 4.2. Assume $\alpha_{\log}(\mathrm{P}^{(T)}) < 1/2$. Then with probability at least $1 - \varepsilon$, ERM $\widehat{\boldsymbol{\theta}}$ satisfies the following:*

$$L(\widehat{\boldsymbol{\theta}}) \le \inf_{\boldsymbol{\theta} \in \Theta_{L,M,D',B}} L(\boldsymbol{\theta}) + O\left(\frac{B_x^2}{1 - \alpha^n(\mathrm{P}^{(T)})}\sqrt{\frac{L(MD^2 + DD')\zeta + \log(1/\varepsilon)}{nT}}\right),$$

*where $C$ is an universal constant, and $\zeta = O(\log(2 + \max\{B, \mathtt{R}, B_x, T, d\}))$.*

The proof is in Appendix E.4. Here we observe that the bound mainly depends on two terms, the Dobrushin's coefficient, and the demoninator $nT$. Under Dobrushin's condition, increasing $n$ alleviates the performance degradation of $1/1-\alpha$ exponentially in $n$. Further, increasing $n$ also tightens the bound with the order $1/\sqrt{n}$. Further, in Theorem 4.4, we do not assume our data is generated from the AR process, only its Dobrushin coefficient. When the data is generated by the AR process, we are able to give a more explicit bound on the same test loss as described below.

**Proposition 4.5** (Test Error Bound). *Following the setup in Theorem 4.4, if pretraining samples are generated by some $\mathtt{AR}_d(q)$ process with noise sampled from $N(0, \sigma_\epsilon^2)$[2], then with probability $\Delta(1 - \varepsilon)$, ERM $\widehat{\boldsymbol{\theta}}$ satisfies the following:*

$$L(\widehat{\boldsymbol{\theta}}) \leq O\left(B_x B_w \exp\left(\frac{-L}{\kappa}\right) + \frac{B_x^2}{1 - \alpha^n(\mathtt{P}^{(T)})} \sqrt{\frac{L(MD^2 + DD')\zeta + \log(1/\varepsilon)}{nT}}\right).$$

*where $\Delta = O\left(1 - (\sigma_\epsilon/B_x B_w e^{-L/2\kappa})^2\right)$, $C$ is an universal constant, and $\zeta = O(\log(2 + \max\{B, \mathtt{R}, B_x, T, d\}))$.*

Considering the model parameters $(M, D, D', d)$ are of constant level, one is able to further optimize the bound to $L(\widehat{\boldsymbol{\theta}}) \lesssim (nT)^{-1/2}$ by selecting $L$ appropriately. Next we provide an example of the application of Proposition 4.5 on AR(1) process with the following form

$$\boldsymbol{x}_{t+1} = \langle \boldsymbol{w}, \boldsymbol{x}_t \rangle + \epsilon_t, \quad \epsilon \sim N(0, \sigma_\epsilon^2), \quad \text{with } \boldsymbol{x}_t \in \mathbb{R}^d, \ \boldsymbol{w} \in \mathbb{R}^d, \ \epsilon \in \mathbb{R}, \ \boldsymbol{y}_{t+1} = \boldsymbol{x}_{t+1}^1.$$

To satisfy the condition of $\alpha(\mathtt{P}) < \frac{1}{2}$, we assume $B_x^2 < \ln 1/2 + (\sigma_\epsilon^2)$, $||\boldsymbol{w}||_\infty < 1$. The first condition comes from the fact that we require the pair-wise potential of this time series to be less than $1/2$ (For more details, see Appendix E.5). The second condition comes from the requirement of it being stationary.

**Corollary 4.6** (Generalization Bound for MOIRAI on AR(1)). *Considering $n$ AR(1) processes with each of its Dobrushin's coefficient bounded by $1/2$. With probability at least $\delta(1 - \varepsilon)$, ERM $\widehat{\boldsymbol{\theta}}$ satisfies the following:*

$$L(\widehat{\boldsymbol{\theta}}) = O\left(\frac{\sigma_\epsilon}{\sqrt{1 - \delta}} + \frac{\sigma_\epsilon^2}{B_x} \exp\left(\frac{-L}{\kappa}\right) + \frac{\sigma_\epsilon^2}{1 - \alpha^n(\mathtt{AR}(1))} \sqrt{\frac{L(MD^2 + DD')\zeta + \log(1/\varepsilon)}{nT}}\right),$$

*with*

$$\alpha(\mathtt{AR}(1)) \leq \min\left\{\frac{1}{2}, \frac{2\sqrt{2/\pi} B_x \|w\|_*}{\sigma_\epsilon}\right\},$$

*and $\zeta = O(\log(2 + \max\{B, \mathtt{R}, B_x, d\}))$. If we further optimize the bound by viewing the hyperparameters as constants, the test error obeys $O(e^{-L} + \sqrt{\frac{L}{nT}})$ w.h.p. whenever $\sigma_\epsilon$ is small.*

**Remark 4.7.** *Corollary 4.6 provides an application of Dobrushin's condition on common statistical models, indicating that Dobrushin's condition can be further used to certify stability and derive generalization bounds for a broad class of time series processes beyond the i.i.d. setting.*

**Setup.** We use MOIRAI-base, a 12 layer MOIRAI transformer. The hyperparameters of this experiment are in Table 2. We use MSE loss for pretraining instead of NLL loss for simplicity. For pretraining, we follow the setup in Woo et al. (2024) but set the patch size as 1 to minimize the impact of patch embedding. During pretraining, each time series is randomly sampled. We evaluate the pretrained model on our test data with $d = \{3, 4, 5\}$, $q = 5$ and $\sigma^2 = 1$ with different input length. We compare MOIRAI with LSR performing different gradient descent steps. For LSR, we assume $q$ is known. When MOIRAI takes a $T$ length input, LSR is trained on $T - 1$ samples with each having $dq$ features. A detailed example on our implementation is in Appendix F.4. We also include the standard transformers and MOIRAI with ReLU replacing Softmax, which we term it as MOIRAI-relu. For standard transformers, we keep the any-variate encoding but use standard attention. [3]Details of data generation can be found in Appendix F.3.

## 5 EXPERIMENTS

To verify our analysis, we first train transformers on synthetic datasets generated from AR process with different parameters. The goal of this experiment is to verify the existence of a transformer

---

[2]Here we assume fixed $d, q$ across all samples as one can describe a lower dimension/order AR process with zero coefficients.

[3]In (Woo et al., 2024), without any-variate attention, the error of MOIRAI-small increases roughly 40%.

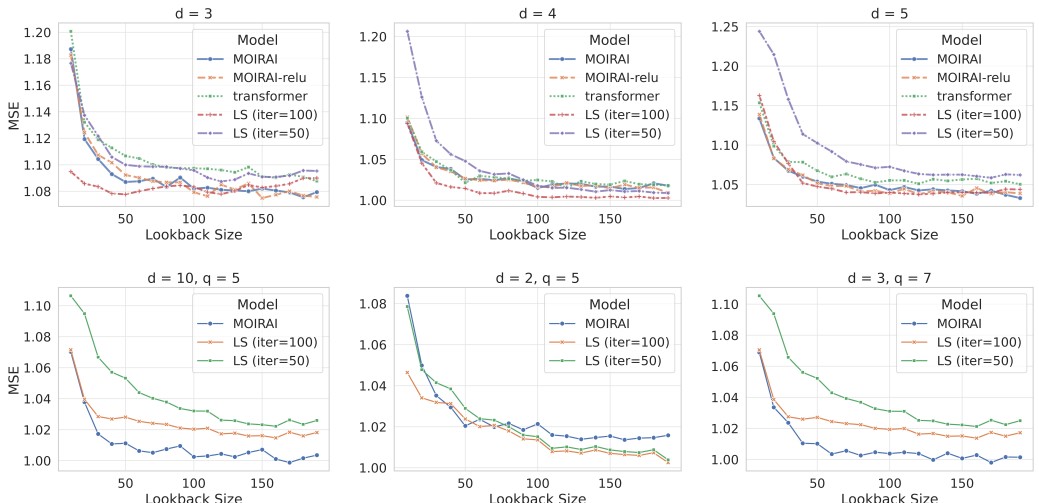

Figure 1: *Top: Model performance on data with different number of covariates.* For both MOIRAI-based models, we observe their performance behave like least squares. As in our construction, the longer the lookback size is, the more examples available for transformers to fit an AR. Note that our test data has variance $\sigma^2 = 1$, thus the MSE for both models are expected to converge to 1 as the $q$ increases. *Bottom: Generalization to unseen values of $d, q$.* From left to right, we have MOIRAI's generalization power (pretrained on $d \in \{4, 5\}, q \in \{4, 5\}$) on high dimensional ($d = 10$), low dimensional data ($d = 2$) and high lag step + low dimensional data ($d = 3, q = 7$). Note that high and low is *compared with pretraining data*. We observe that even when MOIRAI did not learn from any time series with $d = 10$, it is still able to generalize well and shows even better sample complexity than LSR. Finally, even when both $q, d$ are unseen, it does not impact MOIRAI's ability to make accuracy predictions. The lines are the average over 5 runs.

that performs least squares regression (LSR) on input time series with bounded lags and number of covariates. Next we study whether a pretrained transformer is capable of generalize such an ability to unseen values of $d, q$. More empirical results such as on *real-world* data are in Appendix F.5.

**Results.** Since our test data generation process obeys noise variance = 1, when fitting a linear model, the expected MSE will converge towards 1 as lookback size increases. Based on Theorem 3.6, the length of input time series also corresponds to the number of examples model perform least squares on via gradient descent. We observe that as the input length increases, the predictive error of MOIRAI decreases similarly to least squares, which verifies Theorem 3.6. Next, when pretrained on diverse dataset, pretrained MOIRAI is able to adapt to different number of covariates and perform least squares accordingly. Further, when replace softmax with ReLU, the performance gap is negligible. For standard transformer, while it also behaves similar to other models, it does present higher error comparing to other baselines, indicating the advantages of using any-variate attention.

Next we are interested in whether a pretrained transformer can generalize to unseen values of $d$ and $q$. Thus, we train transformers (MOIRAI) on synthetic data generated with AR with $d \in \{4, 5\}$, and $q \in \{4, 5\}$. In our construction, pretrained transformer is compatible with lower order and dimension AR data. We evaluate the trained model on data with unseen values of $d$. We select $d = 2, d = 10$, to represent the scenario when the number of covariates is lower and higher than pretraining data.

**Results.** We observe that even when facing data with unseen number of covariates, MOIRAI is still capable of performing AR regression effectively. Note that for $d = 10$, LS require higher sample complexity to obtain similar performance to $d = 5$ case. However, the pretrained MOIRAI is able to outperform it from such an aspect. For $d = 2$ all models perform well, again verifies our theoretical results. Finally, when facing data with unseen both $d$ and $q$, it is still capable of performing well.

## 6 DISCUSSION

Here we discuss the implications of our results. First, we demonstrate how transformers perform univariate AR regression, a classic time series prediction algorithm. Next, we demonstrate that the unique any-variate attention proposed by Woo et al. (2024), indeed provides benefits for transformers to process each covariate in a parallel fashion (Lemma 3.4). This observation not only leads to

our next result, showing MOIRAI can perform AR regression on arbitrary number of covariates, also provides a justification on the design of any-variate encoding (Woo et al., 2024). The above approximation results provide profound insight on designing transformer-based time series models. Next, for pretraining, we derive generalization bounds when data is under Dobrushin's condition, where the bound scales roughly by $\frac{1}{\sqrt{nT}}$. It is worth investigating how to further relax this assumption, which we leave to future work. For predicting multiple time steps or multiple variates, one can easily apply Bai et al. (2024, Proposition A.5) to obtain the same result with hidden dimension $D$ scales linearly. More discussions on the limitations of this work is in Appendix B.

**Which transformers do we cover?** Our results in Section 3.1 also covers notable SOTA models including classic transformers (Vaswani et al., 2017), Chronos (Ansari et al., 2024) (Theorem E.5), MOMENT (Goswami et al., 2024). Architectures using transformer-decoders are also compatible, such as TimeGPT (Garza and Mergenthaler-Canseco, 2023), TimesFM (Das et al., 2023), see Appendix B in (Bai et al., 2024). Furthermore, the use of patch embedding Nie et al. (2022) does not affect our theoretical result as an AR sequence remains an AR sequence after patch embedding transformation (linear), thus solvable by our AR regression framework. One simply needs to use a weight-tying trick (for the last layer embedding), to obtain the prediction of the original input. Univariate models with patch embedding also fits Section 3.1.

Notable transformer-based SOTA models we do not cover include iTransformer (Liu et al., 2023), CrossFormer (Zhang and Yan, 2023), as they also applied attention mechanism on the covariate dimension. This novel technique requires further analysis thus we leave to future work. Autoformer (Wu et al., 2021) also utilizes a technique named auto-correlation, which requires different theoretical tool to study. Similarly, Time-LLM (Jin et al., 2023) and Lag-Llama (Rasul et al., 2023) utilize either natural language inputs, or the lag features, therefore cannot be covered by our theoretical analysis.

**Can we verify Dobrushin's condition on time series data?** Dobrushin's condition is a property of the joint distribution, not of the observed time series values alone. Therefore, one in general cannot verify this condition without knowing the true distribution behind the data. However, a practical solution we suggest is to first fit commonly used models on the observed time series such as AR, ARMA etc, and then verify the condition on the fitted model. The sufficient conditions for several models to satisfy Dobrushin's condition is derived in Appendix E.5. We also provide a python code that automatically test input time series with the approach described above. Note that it is possible to switch from Dobrushin's condition to $\beta$-mixing Mohri and Rostamizadeh (2010), a more strict condition on non IID data, in our generalization analysis. By doing so, there existing an algorithm Mcdonald et al. (2011) to estimate the $\beta$-mixing coefficient via histograms. Finally, the authors wish to highlight that those dependency assumptions are mainly used for theoretical studies as they excluded many unlearnable examples. These assumptions do not imply they are necessary to the success of model training.

ACKNOWLEDGMENTS

DW is supported by the Northwestern Advanced Cognitive Science Fellowship. YC is supported in part by NSFC 12301657, Hong Kong RGC ECS 27308624, and Hong Kong RGC GRF 17301825. JF research was supported in part by the NSF Grants DMS-2210833 and DMS-2412029 and the ONR Grant N00014-25-1-2317 HL is partially supported by NIH R01LM1372201, NSF AST-2421845, Simons Foundation MPS-AI-00010513, AbbVie, Dolby and Chan Zuckerberg Biohub Chicago Spoke Award. This research was supported in part through the computational resources and staff contributions provided for the Quest high performance computing facility at Northwestern University which is jointly supported by the Office of the Provost, the Office for Research, and Northwestern University Information Technology.

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

# SUPPLEMENTARY MATERIAL

## A  TABLE OF NOTATIONS

Table 1: Mathematical Notations and Symbols

| Symbol | Description |
|---|---|
| $\boldsymbol{x}_i$ | The $i$-th component of vector $\boldsymbol{x}$ |
| $\langle \boldsymbol{a}, \boldsymbol{b} \rangle$ | Inner product for vectors $\boldsymbol{a}, \boldsymbol{b} \in \mathbb{R}^d$ |
| $[I]$ | Index set $\{1, \cdots, I\}$, where $I \in \mathbb{N}^+$ |
| $\|\cdot\|$ | Spectral norm, equivalent to the $l_2$-norm when applied to a vector |
| $\|\cdot\|_{2,\infty}$ | The largest L2 norm of column vectors of a matrix |
| $\boldsymbol{A}_{ij}$ | The element on the $i$-th row and $j$-th column of matrix $\boldsymbol{A}$ |
| $\boldsymbol{x}_{i:j}$ | The sub-sequence of sequence $\boldsymbol{x}$ from coordinate $i$ to $j$ |
| $\oplus$ | Concatenation between column vectors $\boldsymbol{v} \oplus \boldsymbol{u} \mapsto (\boldsymbol{v}^\top, \boldsymbol{u}^\top)^\top$ |
| $[\boldsymbol{u}; \boldsymbol{v}]$ | Concatenation between two row vectors |
| $N$ | Length of a transformer input sequence |
| $T$ | Number of time steps of a time series |
| $M$ | Number of attention heads. |
| $q$ | Lag of an AR process. |
| $d$ | The number of covariates in an AR process |
| $\boldsymbol{v}$ | Vector (bold lower) |
| $\boldsymbol{A}$ | Matrix (bold upper) |
| $\mathcal{X}$ | random variable (calligraphic) |
| x | element from a domain set |
| $\mathcal{D}_\mathcal{X}$ | Domain of random variable $\mathcal{X}$ |
| $\boldsymbol{e}_i$ | one-hot vector with its $i$-th entry as 1 |
| $\mathsf{P}_\mathcal{X}$ | Probability distribution of $\mathcal{X}$ |
| $P_{\boldsymbol{z}|\boldsymbol{w}}(z \mid w)$ | The probability $P[\boldsymbol{z} = z \mid \boldsymbol{w} = w]$ |

## B    LIMITATIONS AND IMPACT

**Limitations.**    One limitation in our analysis is that we consider ReLU instead of softmax in attention mechanisms. While the same approach also is in theoretical (5; 22; 15) and empirical works (41; 43; 33), one might obtain a different approximation bound comparing to Theorem 3.6. However, in our generalization analysis, the difference is small as softmax does not affect the model complexity too much. Another aspect is that we mainly focus on AR processes. While in the appendix, we do show the approximation result for non-linear AR processes generated by a ReLU network, to achieve universal forecasting, a more general assumption on data is required. Another limitation is we consider MSE loss instead of NLL loss used in (40) for pretraining.

**Impact Statement.**    This paper studies the theoretical aspect of transformers as time series foundation models. No negative societal impacts that the authors feel should be specifically highlighted here.

## C    RELATED WORKS

**Transformers in Time Series Forecasting.**    The recent progress in foundation models (35; 6; 29) has begun to reshape the field of time series forecasting, a critical task of predicting the future based on history (14). However, there are two major challenges in building a time series foundation model: (a) the model must be able to handle an arbitrary number of covariates, and (b) the model must generalize to unseen time series domains. To circumvent (a), several studies simplify the task by considering only univariate time series (4; 30; 8). (8) propose a decoder-only transformer pretrained on both real and synthetic datasets. (30) incorporate lag features and the Llama architecture to pretrain a large univariate time series foundation model. (4) leverage the power of large language models (LLMs) by using pretrained LLMs backbones.

Recently, (40) proposed MOIRAI, the first time series foundation model capable of handling an arbitrary number of covariates. It addresses (a) by concatenating all covariates into a uni-dimensional sequence, ensuring a consistent input dimension across datasets. It addresses (b) by pretraining on a large collection of time series datasets (12; 3; 42; 19) spanning domains such as weather, traffic, electricity, and industry. MOIRAI not only generalizes across a wide range of domains, but its *zero-shot* performance also surpasses several strong supervised learning baselines (23; 28; 45). However, the machine learning community has yet to provide a suitable explanation for MOIRAI's impressive performance. Therefore, this paper is the first to offer theoretical guarantees for MOIRAI as a time series foundation model.

**In-Context Learning.**    In-context learning (ICL) is an emerging capability of large foundation models, enabling them to learn diverse and unseen tasks from given examples. (6) first provide empirical evidence of ICL in large language models (LLMs); by presenting several examples of $(\boldsymbol{x}, \boldsymbol{y})$ pairs, GPT-3 effectively infers the relationship between $\boldsymbol{x}$ and $\boldsymbol{y}$. (10) then conduct quantitative experiments on simple function classes, such as linear regression. Their results demonstrate that large foundation models can learn the parameters of these function classes. Subsequently, several theoretical studies (5; 37; 1; 2) have proven that different types of transformers can implement algorithms such as gradient descent. This discovery provides a theoretical foundation for the empirical findings in (10).

The closest studies to this paper are (27; 32). However, (32) examines ICL in the context of next-token prediction using a linear transformer. While their theoretical results relate to in-context learning on sequential data, they are insufficient to explain transformers' success in time series forecasting. (27) explores another case where the data is modeled as a Markov chain generated by a transition matrix. They demonstrate the existence of induction heads that enable transformers to perform next-token prediction. However, their scenario does not align with multivariate time series, which is where our main contribution lies.

The biggest different between our work to existing ICL studies (5; 27; 37; 24) is we do not consider structured inputs as in practice, time series data is considered a long sequence instead of many feature-label pairs. Next, in our case study, we show that MOIRAI can perform AR regression on different dimensions of feature, while ICL studies mostly focused on a fixed dimension of features. Last, our pretraining analysis considers the case where data is non IID, which is critical in time series data due to the dependent nature. Therefore, this paper indeed provides new insights on using transformers for time series learning. Another related work on both time series analysis, in-context

learning and transformers is (46). In our paper, we study the function class of MOIRAI, where (46) studies transformers with LSA with no-MLPs. Next, while in our paper, our construction also reshapes input sequence into Hankel-like matrices (sliding window features) and then performs LSR by gradient descent. On the other hand, (46) directly feeds the Hankel matrix $H$ as input and shows that the sigma-algebra generated by $LSA(H)$ is strictly coarser than generated by $H$ itself, so LSA inevitably loses information relative to OLS. Also, our construction is algorithmic, and (46) focuses on representation space, and our results rely on two key powers that (46) does not have (Lemma 3.2, 3.4, Prop 3.3, Thm 3.6). Further, our construction uses model depth to unroll GD steps and (46) uses model depth to compose cubic maps, which cannot simulate least square regression. Thus, the difference between our work and (46) lies in our settings and goals. Finally, we have added a short discussion on this work in the related work section.

**Learning Theory on Time Series Data.** Efforts in time series learning theory have developed frameworks for non-i.i.d. data using mixing or ergodic assumptions (7; 47; 26; 34). Yet these results are largely agnostic to model architecture and thus offer limited insight into the specific advantages of transformers. In short, a clear theoretical understanding of how transformer architectures contributes to time series forecasting remains elusive.

**Relationships to (5).** Here we first summarize the contribution of (5). Their major contribution is to demonstrate there exists a transformer to perform gradient descent in-context, which provides a great starting point to provide possibile explanation on the empirical success of transformers in ICL. Their generalization results focus on pretraining on ICL tasks. However, they mainly focus on the standard transformer architecture (with ReLU activation) and IID samples. They did not cover:

    A  Arbitrary number of covariates (feature dimension in their paper)

    B  Any-variate encoding (different input format)

    C  Any-variate attention (MOIRAI's new architecture)

    D  Non-IID learning on time series

On the other hand, we focus on MOIRAI, a specific architecture and encoding method designed to address the problem where standard transformers cannot handle an arbitrary number of covariates. Their design/approach is simple but powerful. We extend the results from Bai et al. (5) to tackle the above 4 setting differences (which is realistic to MOIRAI) with our novel results. To address (B), we use our Lemma 3.2, to deomnstrate how one can apply Bai et al. (5)'s results to MOIRAI with minimal effort. For (A, C), we provide Lemma 3.4, which demonstrates a key insight on how any-variate attention can handle (A, C). Both Lemma 3.2 and Lemma 3.4 are novel results and not based on Bai et al. (5). Finally, for non-IID time series pretraining, we introduce Dobrushin's condition and new proving techniques (Corollary E.16, Lemma E.18) to obtain generalization on time series learning.

## D   ADDITIONAL THEORETICAL BACKGROUND

Here, we include several technical lemme that are intensively used throughout our paper. The Lipschitzness of an MLP layer is obtained in (5, Lemma J.1), which we restate it below

**Lemma D.1** ((5)). *For a single MLP layer,* $\boldsymbol{\theta}_2 = (\boldsymbol{W}_1, \boldsymbol{W}_2)$*, we introduce its norm*

$$|||\boldsymbol{\theta}_2||| = ||\boldsymbol{W}_1||_{op} + ||\boldsymbol{W}_2||_{op}.$$

*For any fixed hidden dimension* $D'$*, we consider*

$$\Theta_{2,B} \coloneqq \{\boldsymbol{\theta}_2 : |||\boldsymbol{\theta}_2||| \leq B\}.$$

*Then for* $\boldsymbol{H} \in \mathcal{H}_R$*,* $\boldsymbol{\theta}_2 \in \Theta_{2,B}$*, the function* $(\boldsymbol{\theta}_2, \boldsymbol{H}) \mapsto MLP_{\boldsymbol{\theta}_2}$ *is* $(BR)$*-Lipschitz w.r.t.* $\boldsymbol{\theta}_2$ *and* $(1 + B^2)$*-Lipschitz w.r.t.* $\boldsymbol{H}$*.*

The following lemma shows any-variate attention is capable of performing variate-wise operation on arbitrary number of covariates under any-variate encoding.

**Lemma D.2** (Group-Wise Operation via Any-Variate Attention). *Let* $||\boldsymbol{H}||_{2,p} \coloneqq (\sum_{i=1}^{N} ||\boldsymbol{h}_i||_2^p)^{1/p}$ *denote the column-wise* $(2, p)$*-norm of* $\boldsymbol{H}$*. For any input matrix* $\boldsymbol{H} = (\boldsymbol{h}_1, \cdots, \boldsymbol{h}_T)$ *such that*

$||\boldsymbol{H}||_{2,\infty} \le$ R, *suppose* $\psi(\cdot) : \mathbb{R}^{D \times T} \to \mathbb{R}^{D \times T}$ *is a sequence-to-sequence function implemented by a single layer standard transformer (TF*$_{\boldsymbol{\theta}}^{\dagger}$*) such that*

$$TF_{\boldsymbol{\theta}}^{\dagger}(\boldsymbol{H}) \coloneqq \psi(\boldsymbol{H}).$$

*Then there exists a single layer MOIRAI transformer TF*$_{\boldsymbol{\theta}}(\cdot)$ *such that for any input*

$$\boldsymbol{H}^{\star} = \begin{bmatrix} \boldsymbol{H}_1 & \boldsymbol{H}_2 & \cdots & \boldsymbol{H}_K \end{bmatrix},$$

*where* $\boldsymbol{H}_k \in \mathbb{R}^{D \times T}$. *TF*$_{\boldsymbol{\theta}}(\cdot)$ *performs*

$$TF_{\boldsymbol{\theta}}(\boldsymbol{H}^{\star}) = \begin{bmatrix} \psi(\boldsymbol{H}_1) & \psi(\boldsymbol{H}_2) & \cdots & \psi(\boldsymbol{H}_K) \end{bmatrix}.$$

*Proof of Lemma D.2.* We start by showing the case of a single-head, single-layer standard transformer. Let

$$\text{MLP}_{\boldsymbol{\theta}_2} \circ \text{Attn}_{\boldsymbol{\theta}}^{\dagger}(\boldsymbol{H}) = \text{MLP}_{\boldsymbol{\theta}_2} \circ \boldsymbol{V}\boldsymbol{H}\sigma(\langle \boldsymbol{Q}\boldsymbol{H}, \boldsymbol{K}\boldsymbol{H}\rangle) = \boldsymbol{V}\boldsymbol{H}\boldsymbol{A}_{\boldsymbol{H}},$$

where $\boldsymbol{A}_{\boldsymbol{H}} = \sigma(\langle \boldsymbol{Q}\boldsymbol{H}, \boldsymbol{K}\boldsymbol{H}\rangle)$.

Let $\psi_1(\boldsymbol{H}) \coloneqq \boldsymbol{V}\boldsymbol{H}\boldsymbol{A}_{\boldsymbol{H}}$, to apply group-wise operation of $\psi_1(\cdot)$ on some input such that

$$\psi_1(\boldsymbol{H}^{\star}) = \begin{bmatrix} \psi_1(\boldsymbol{H}_1) & \psi_1(\boldsymbol{H}_2) & \cdots & \psi_1(\boldsymbol{H}_K) \end{bmatrix}.$$

Let $\boldsymbol{0} \in \mathbb{R}^{T \times T}$ be a zero matrix, and $\boldsymbol{1} \in \mathbb{R}^{T \times T}$ be a 1s matrix, for for any input $||\boldsymbol{H}^{\star}||_{2,\infty} \le$ R, one can find some $u^2 < 0$ to decompose $\psi_1(\cdot)$ into the following form.

$$\psi_1(\boldsymbol{H}^{\star}) = \boldsymbol{V} \begin{bmatrix} \boldsymbol{H}_1 \boldsymbol{A}_{\boldsymbol{H}_1} & \boldsymbol{H}_2 \boldsymbol{A}_{\boldsymbol{H}_2} & \cdots & \boldsymbol{H}_K \boldsymbol{A}_{\boldsymbol{H}_K} \end{bmatrix}$$

$$= \boldsymbol{V}\boldsymbol{H}^{\star} \begin{bmatrix} \boldsymbol{A}_{\boldsymbol{H}_1} & \boldsymbol{0} & \cdots & \boldsymbol{0} \\ \boldsymbol{0} & \boldsymbol{A}_{\boldsymbol{H}_2} & \cdots & \boldsymbol{0} \\ \vdots & \vdots & \ddots & \vdots \\ \boldsymbol{0} & \boldsymbol{0} & \cdots & \boldsymbol{A}_{\boldsymbol{H}_K} \end{bmatrix}$$

$$= \boldsymbol{V}\boldsymbol{H}^{\star} \times$$

$$\sigma\left( \begin{bmatrix} \langle \boldsymbol{Q}\boldsymbol{H}_1, \boldsymbol{K}\boldsymbol{H}_1\rangle & \langle \boldsymbol{Q}\boldsymbol{H}_1, \boldsymbol{K}\boldsymbol{H}_2\rangle & \cdots & \langle \boldsymbol{Q}\boldsymbol{H}_1, \boldsymbol{K}\boldsymbol{H}_K\rangle \\ \langle \boldsymbol{Q}\boldsymbol{H}_2, \boldsymbol{K}\boldsymbol{H}_1\rangle & \langle \boldsymbol{Q}\boldsymbol{H}_2, \boldsymbol{K}\boldsymbol{H}_2\rangle & \cdots & \langle \boldsymbol{Q}\boldsymbol{H}_2, \boldsymbol{K}\boldsymbol{H}_K\rangle \\ \vdots & \vdots & \ddots & \vdots \\ \langle \boldsymbol{Q}\boldsymbol{H}_K, \boldsymbol{K}\boldsymbol{H}_1\rangle & \langle \boldsymbol{Q}\boldsymbol{H}_K, \boldsymbol{K}\boldsymbol{H}_2\rangle & \cdots & \langle \boldsymbol{Q}\boldsymbol{H}_K, \boldsymbol{K}\boldsymbol{H}_K\rangle \end{bmatrix} + \begin{bmatrix} \boldsymbol{0} & u^2 \cdot \boldsymbol{1} & \cdots & u^2 \cdot \boldsymbol{1} \\ u^2 \cdot \boldsymbol{1} & \boldsymbol{0} & \cdots & u^2 \cdot \boldsymbol{1} \\ \vdots & \vdots & \ddots & \vdots \\ u^2 \cdot \boldsymbol{1} & u^2 \cdot \boldsymbol{1} & \cdots & \boldsymbol{0} \end{bmatrix} \right).$$

Further, observe that operations in an MLP layer are either left multiplication or element-wise operations, which implies group-wise as well. We then finish the proof by setting $u^1 = 0$.

$\square$

**Theorem D.3** ((38, Section 5.6)). *Suppose* $\psi : [0, +\infty \to [0, +\infty)$ *is a convex, non-decreasing function satisfying* $\psi(x + y) \ge \psi(x)\psi(y)$. *For any random variable* $X$, *we consider the Orlicz norm induced by* $\psi : ||X||_{\psi} \coloneqq \inf\{K > 0 : \mathbb{E}\psi(|X|/K)\} \le 1$. *Suppose that* $\{X_{\theta}\}$ *is a zero-mean random process indexed by* $\theta \in \Theta$ *such that* $||X_{\theta} - X_{\theta'}|| \le \rho(\theta, \theta')$ *for some metric* $\rho$ *on* $\Theta$. *Then the following holds*

$$P\left( \sup_{\theta, \theta' \in \Theta} |X_{\theta} - X_{\theta'}| \le 8(J + t) \right) \le \frac{1}{\psi(t/D)}, \quad \text{for all} t \ge 0,$$

*where* $D$ *is the diameter of the metric space* $(\Theta, \rho)$, *and the generalized Dudley entropy integral* $J$ *is given by*

$$J \coloneqq \int_0^D \psi^{-1}(N(\delta; \Theta, \rho))d\delta,$$

*where* $N(\delta; \Theta, \rho)$ *is the* $\delta$-*covering number of* $(\Theta, \rho)$.

The next technical lemma is in (5). Let $\text{B}_{\infty}^k(R) = [-R, R]^k$ denotes the standard $\ell_{\infty}$ ball in $\mathbb{R}^k$ with radius $R > 0$.

**Definition D.4** (Sufficiently smooth $k$-variable function). *We say a function $g : \mathbb{R}^k \mapsto \mathbb{R}$ is $(R, C_\ell)$-smooth if for $s = \lceil (k-1)/2 \rceil + 2$, $g$ is a $C^s$ fnction on $\mathtt{B}_\infty^k(R)$, and*

$$\sup_{\boldsymbol{z} \in \mathtt{B}_\infty^k(R)} \left\| \nabla^i g(\boldsymbol{z}) \right\|_\infty = \sup_{\boldsymbol{z} \in \mathtt{B}_\infty^k(R)} \sup_{j_1, \cdots, j_i \in [k]} |\partial_{x_{j1} \cdots x_{ji}} g(\boldsymbol{x}) \leq L_i$$

*for all $i = 0, 1, \cdots, s$, with $\max_{0 \leq i \leq s} L_i R^i \leq C_\ell$.*

**Lemma D.5** (Approximating smooth $k$-variable functions). *For any $\varepsilon_{approx} > 0$, $R \geq 1$, $C_\ell > 0$, we have the following: Any $(R, C_\ell)$-smooth function $g : \mathbb{R}^k \mapsto \mathbb{R}$ is $(\varepsilon_{approx}, R, M, C)$-approximable by sum of relus with $M \leq C(k) C_\ell^2 \log(1 + C_\ell / \varepsilon_{approx}^2)$ and $C \leq C(k) C_\ell$, where $C(k) > 0$ is a constant that depends only on $k$, i.e.,*

$$f(\boldsymbol{z}) = \sum_{m=1}^M c_m \sigma(\boldsymbol{a}_m^\top [\boldsymbol{z}; 1]) \quad \text{with } \sum_{m=1}^M |c_m| \leq C, \quad \max_{m \in [M]} \|\boldsymbol{a}_m\|_1 \leq,$$

*such that $\sup_{\boldsymbol{z} \in [-R, R]^k} |f(\boldsymbol{z}) - g(\boldsymbol{z})| \leq \varepsilon_{approx}$.*

# E  PROOFS

## E.1  PROOF OF LEMMA 3.2

Here we prove a slightly simpler result with the positional encoding containing only zero vectors and a one-hot vector. One can easily extend the proof by padding the weight matrices.

$$\boldsymbol{H} := \begin{bmatrix} x_1 & x_2 & \dots & x_T & 0 \\ \boldsymbol{p}_1 & \boldsymbol{p}_2 & \dots & \boldsymbol{p}_T & \boldsymbol{p}_{T+1} \end{bmatrix} \in \mathbb{R}^{D \times (T+1)}, \quad \boldsymbol{p}_i := \begin{bmatrix} \boldsymbol{0}_{d'} \\ \boldsymbol{e}_i \end{bmatrix} \in \mathbb{R}^{d'+T}, \tag{E.1}$$

**Lemma E.1** (Lemma 3.2 Restate). *Given a sequence of token $\boldsymbol{H}$ in the form of Equation E.1, there exists a one-layer, $q - 1$ head ReLU attention layer, such that the columns of $Attn_{\boldsymbol{\theta}}(\boldsymbol{H})$ has the following form:*

$$Attn^{\dagger}_{\boldsymbol{\theta}_1}(\boldsymbol{H})_i := \begin{bmatrix} x_i \\ x_{i-1} \\ \vdots \\ x_{i-q} \\ \boldsymbol{p}'_i \end{bmatrix}, \quad where \ \boldsymbol{p}'_i := \begin{bmatrix} \boldsymbol{0}_{d'-q} \\ 1 \\ 1\{i < T+1\} \end{bmatrix} \in \mathbb{R}^{d'-q+2}. \tag{E.2}$$

*Proof.* Consider an input of the following form

$$\boldsymbol{x} = \begin{bmatrix} x_1 & x_2 & \cdots & x_T \\ \boldsymbol{0} & \boldsymbol{0} & \cdots & \boldsymbol{0} \\ \boldsymbol{e}_1 & \boldsymbol{e}_2 & \cdots & \boldsymbol{e}_T \end{bmatrix},$$

where $\boldsymbol{x}_t \in \mathbb{R}^d, \mathbf{p}_t \in \mathbb{R}^T$, for all $t = 1, \cdots, T$. We construct weights of the $m$-th head $\boldsymbol{W}^m_Q, \boldsymbol{W}^m_Q$ as following,

$$\boldsymbol{W}^m_K = \begin{bmatrix} \boldsymbol{0}^{\top} & \boldsymbol{0}^{\top} & \boldsymbol{e}^{\top}_1 \\ \boldsymbol{0}^{\top} & \boldsymbol{0}^{\top} & \boldsymbol{e}^{\top}_2 \\ \vdots & \vdots & \vdots \\ \boldsymbol{0}^{\top} & \boldsymbol{0}^{\top} & \boldsymbol{e}^{\top}_T \end{bmatrix}, \quad \boldsymbol{W}^m_Q = \begin{bmatrix} \boldsymbol{0}^{\top} & \boldsymbol{0}^{\top} & \boldsymbol{e}^{\top}_{1-m} \\ \boldsymbol{0}^{\top} & \boldsymbol{0}^{\top} & \boldsymbol{e}^{\top}_{2-m} \\ \vdots & \vdots & \vdots \\ \boldsymbol{0}^{\top} & \boldsymbol{0}^{\top} & \boldsymbol{e}^{\top}_{T-m} \end{bmatrix},$$

where we define the negative index as rotational index, i.e., $\boldsymbol{e}_{-1} = \boldsymbol{e}_T, \boldsymbol{e}_{-2} = \boldsymbol{e}_{T-1}$. We have

$$(\boldsymbol{W}^m_K \mathbf{X})^{\top} (\boldsymbol{W}^m_Q \mathbf{X}) = \begin{bmatrix} \boldsymbol{e}^{\top}_1 \\ \boldsymbol{e}^{\top}_2 \\ \vdots \\ \boldsymbol{e}^{\top}_T \end{bmatrix}^{\top} \begin{bmatrix} \boldsymbol{e}^{\top}_{1-m} \\ \boldsymbol{e}^{\top}_{2-m} \\ \vdots \\ \boldsymbol{e}^{\top}_{T-m} \end{bmatrix}$$

$$= \boldsymbol{I}_T \begin{bmatrix} \boldsymbol{e}_{1-m} \\ \boldsymbol{e}_{2-m} \\ \vdots \\ \boldsymbol{e}_{T-m} \end{bmatrix}.$$

Note that the result of $\sigma\left((\boldsymbol{W}^m_K \mathbf{X})^{\top} (\boldsymbol{W}^m_Q \mathbf{X})\right)$ is a rotation matrix, where right multiplication on $\mathbf{X}$ will rotate the columns of $\mathbf{X}$. Therefore, we have $\boldsymbol{W}^m_V$ that performs row-wise shifting and the attention matrix $\sigma\left((\boldsymbol{W}^m_K \mathbf{X})^{\top} (\boldsymbol{W}^m_Q \mathbf{X})\right)$ performs column-wise shifting. $\square$

## E.2  PROOF OF THEOREM 3.6

**Autoregressive Linear Regression under Any-Variate Encoding.**   The ultimate goal of this setup is to perform the following mechanism. Let $\boldsymbol{x}$ be the target variate we wish to predict, $\boldsymbol{z}^j$ be the $j$-th covariate of $\boldsymbol{x}$, for $j \in [M]$. We denote the lookback window size as $q$, and each covariate has length $T$ ($T$-time steps.). We denote the time encoding as $\boldsymbol{p}_i$ for $i \in [T]$, and the variate encoding as $\boldsymbol{q}_j$ for $j \in [M]$. Finally, our goal is to predict $\boldsymbol{x}_T$.

$$
\begin{bmatrix}
x_1^1 & \cdots & x_T^1 & x_1^2 & \cdots & x_T^2 & \cdots & x_1^d & \cdots & x_T^d \\
\boldsymbol{p}_1 & \cdots & \boldsymbol{p}_T & \boldsymbol{p}_1 & \cdots & \boldsymbol{p}_T & \cdots & \boldsymbol{p}_1 & \cdots & \boldsymbol{p}_T \\
\boldsymbol{e}_1 & \cdots & \boldsymbol{e}_1 & \boldsymbol{e}_2 & \cdots & \boldsymbol{e}_2 & \cdots & \boldsymbol{e}_d & \cdots & \boldsymbol{e}_d
\end{bmatrix}
\mapsto
\begin{bmatrix}
\mathtt{A}_1(q) & \cdots \\
\mathtt{A}_2(q) & \cdots \\
\vdots & \\
\mathtt{A}_d(q) & \cdots \\
\vdots & \ddots
\end{bmatrix}.
$$

Here, different colors represent different covariates. The motivation for performing such an operation is to apply the in-context learning property of transformers proved in (5).

**Lemma E.2** (Lemma 3.4 Restate). *Define the matrix $\mathtt{A}_i(q)$ for the $i$-th covariates $(x_1^i, \cdots, x_T^i)$, with order $q$, such that*

$$
\mathtt{A}_i(q) :=
\begin{bmatrix}
x_1^i & x_2^i & \cdots & x_t^i & x_{t+1}^i & x_{t+2}^i & \cdots \\
x_T^i & x_{T-1}^i & \cdots & x_{t-1}^i & x_t^i & x_{t+1}^i & \cdots \\
x_{T-1}^i & x_{T-2}^i & \cdots & x_{t-2}^i & x_{t-1}^i & x_t^i & \cdots \\
\vdots & \vdots & \vdots & \vdots & \vdots & \vdots & \cdots \\
x_{T-q}^i & x_{T-q+1}^i & \cdots & x_{t-q}^i & x_{t-q+1}^i & x_{t-q+2}^i & \cdots
\end{bmatrix},
$$

*where in the $j$-th column of $\mathtt{A}_i(q)$, it contains historical values of $x_j^i$ with lag $q$.*

*Given fixed $D, T \in \mathbb{N}^+$, where $T > q$. For any input matrix $\boldsymbol{H}$ in the form of Any-Variate Encoding in Equation 3.4, such that $\boldsymbol{H} \in \mathbb{R}^{D' \times dT'}$, and $D' \le D$, $T' < T$. There exists a 1-layer, $q$ head Any-Variate Attention that performs the following operation.*

$$
\begin{bmatrix}
x_1^1 & \cdots & x_T^1 & x_1^2 & \cdots & x_T^2 & \cdots & x_1^d & \cdots & x_T^d \\
\boldsymbol{p}_1 & \cdots & \boldsymbol{p}_T & \boldsymbol{p}_1 & \cdots & \boldsymbol{p}_T & \cdots & \boldsymbol{p}_1 & \cdots & \boldsymbol{p}_T \\
\boldsymbol{e}_1 & \cdots & \boldsymbol{e}_1 & \boldsymbol{e}_2 & \cdots & \boldsymbol{e}_2 & \cdots & \boldsymbol{e}_d & \cdots & \boldsymbol{e}_d
\end{bmatrix}
\mapsto
\begin{bmatrix}
\mathtt{A}_1(q) & \mathtt{A}_2(q) & \cdots & \mathtt{A}_d(q) \\
\ddots & \ddots & \cdots & \ddots
\end{bmatrix}
$$

*Proof.* The proof comes as a direct corollary of Lemma D.2 and (5, Proposition A.5). By Lemma 3.2, there exists a single layer standard transformer layer (with $\boldsymbol{W}_1, \boldsymbol{W}_2$ being 0s) that generates $\mathtt{A}_i(q)$ for each univariate (covariate). It then left applying Lemma D.2 for variate-wise operation and applying (5, Proposition A.5) to keep the time indices $\boldsymbol{p}_t$ unchanged. □

**Corollary E.3.** *There exists a $d_{\max}$ head standard attention layer that performs the following*

$$
\begin{bmatrix}
\mathtt{A}_1(q) & \mathtt{A}_2(q) & \cdots & \mathtt{A}_d(q) \\
\ddots & \ddots & \cdots & \ddots
\end{bmatrix}
\mapsto
\begin{bmatrix}
\mathtt{A}_1(q) & \cdots \\
\widetilde{\mathtt{A}}_2(q) & \cdots \\
\vdots & \\
\widetilde{\mathtt{A}}_d(q) & \cdots \\
\ddots & \ddots
\end{bmatrix}, \quad \text{for any } d \le d_{\max},
$$

*where $\widetilde{\mathtt{A}}_i(q)$ is $\mathtt{A}_i(q)$ without the first row.*

*Proof.* Note that this operation in Corollary E.3 is straightforward with Lemma 3.2 and (5, Proposition A.5). As for each $i \in [d]$, $i \ne 1$, the attention layer performs two operations to each element of $\mathtt{A}_i(q)$:

$$
\begin{cases}
iT \text{ columns to the left} & \text{right multiplication} \\
q_{\max} \text{ rows below} & \text{left multiplication} \\
\text{zero out} & \text{if in first row (left multiplication)}
\end{cases}.
$$

Note that one can simply construct weight matrices to perform the above permutations and masking. In total, we need $d_{\max}$ heads to perform such operations for each $\mathtt{A}_i(q)$, for any $d \le d_{\max}$. For

$q < q_{\max}$, the remaining entries will be zero padded. Finally, with at best 2 layers of $d_{\max}$ head any-variate attention, we then obtain

$$\widetilde{H}^{(2)} := \begin{bmatrix} \widetilde{\mathtt{A}}_1(q) & \cdots \\ \widetilde{\mathtt{A}}_2(q) & \cdots \\ \vdots & \\ \widetilde{\mathtt{A}}_d(q) & \cdots \\ \boldsymbol{p} & \\ \boldsymbol{e} & \end{bmatrix} = \begin{bmatrix} \cdots & x_{T-1}^1 & x_T^1 & \cdots \\ \cdots & x_{T-2}^1 & x_{T-1}^1 & \cdots \\ \vdots & \vdots & \vdots & \cdots \\ \cdots & x_{T-q}^1 & x_{T-q}^1 & \cdots \\ \cdots & x_{T-1}^d & x_{T-1}^d & \\ \cdots & x_{T-2}^d & x_{T-2}^d & \\ \cdots & \vdots & \vdots & \\ \cdots & x_{T-q}^d & x_{T-q}^d & \\ \cdots & \boldsymbol{p}_{T-1} & \boldsymbol{p}_T & \\ \cdots & \boldsymbol{e}_1 & \boldsymbol{e}_1 & \end{bmatrix},$$

where $\boldsymbol{p}$ is the matrix of $(\boldsymbol{p}_1, \cdots, \boldsymbol{p}_T)$, $\boldsymbol{e}$ is the matrix of $(\boldsymbol{e}_1, \cdots, \boldsymbol{e}_1)$.

Note that $x_T^1$ in red is the target we wish to predict (masked as 0 initially), and the entries in blue is considered the input feature of our AR model (a linear regression model in this case), and we are able to directly apply several theoretical results in (5) with input $\widetilde{\boldsymbol{H}}^{(2)}$. Specifically, for Theorem 3.6, it follows directly from (5, Theorem 4) by setting $\lambda = 0$.

$\square$

Next, we present several approximation results from (5), which our approximation results follows immediately from. Considering the general form of autoregressive data: $\boldsymbol{x} \in \mathbb{R}^{d \times T} := (\boldsymbol{x}_1, \ldots, \boldsymbol{x}_T)$, where $\boldsymbol{x}_t = (x_t^1, \cdots, x_t^d) \in \mathbb{R}^d$. Assuming our target (variate of interest) is in dimension 1, we assume the autoregressive process generates $x_t^1$ as follows:

$$x_t^1 = f(\boldsymbol{x}_{t-q:t-1}^{1:d}) + \epsilon_t, \tag{E.3}$$

where $\epsilon_t \sim N(0, \sigma^2)$, $a_i^j \in \mathbb{R}^1$, and $f$ is a function of interest. We then present several results when $f$ varies.

**Non-Linear AR.** Here we analyze that when the autoregressive process is generated by a 2 layer ReLU network with look back window size $q$. Suppose the prediction function $\text{pred}(\boldsymbol{x}, \boldsymbol{w}) := \sum_{k=1}^K u_k r(\boldsymbol{v}_k^\top \boldsymbol{x})$ is given by a two-layer neural network, parameterized by $\boldsymbol{w} = [\boldsymbol{w}_k, u_k]_{k \in [K]} \in \mathbb{R}^{K(d+1)}$. Consider the ERM problem:

$$\min_{\boldsymbol{w} \in \mathcal{W}} \widehat{L}_N(\boldsymbol{w}) := \frac{1}{2N} \sum_{i=1}^N \ell(\text{pred}(\boldsymbol{x}_i, \boldsymbol{w}), y_i) = \frac{1}{2N} \sum_{i=1}^N \ell\left(\sum_{k=1}^K u_k r(\boldsymbol{v}_k^\top \widetilde{\boldsymbol{x}}_i), x_T^1\right),$$

where $\mathcal{W}$ is a bounded domain and $\widetilde{\boldsymbol{x}}_i \in \mathbb{R}^{qd}$ is a flatten version of $\boldsymbol{x}_{t-q:t-q} \in \mathbb{R}^{d \times q}$.

**Proposition E.4.** *Fix any $B_w, B_u > 0$, $L \geq 3, \nu > 0$, and $\varepsilon > 0$. Suppose that*

1. *Both the activation function $r$ and the loss function $\ell$ is $C^4$-smooth.*

2. *$\mathcal{W}$ is a closed domain such that $\mathcal{W} \subset \{\boldsymbol{w} = [\boldsymbol{v}_k; u_k]_{k \in [K]} \in \mathbb{R}^{K(d+1)} : ||\boldsymbol{v}_k||_2 \leq B_v, |u_K| \leq B_u\}$, and $\text{Proj}_{\mathcal{W}} = MLP_{\boldsymbol{\theta}_2}$ for some MLP layer with hidden dimension $D_w$ and $||\boldsymbol{\theta}_2||_{op} \leq C_w$.*

*Then there exists a $(L_1 + 2L_2)$-layer MOIRAI transformer with*

$$\max_{\ell \in [L_1+1, 2L_2]} M^{(\ell)} \leq \widetilde{O}(\varepsilon^{-2}), \quad \max_{\ell \in [L_1+1, 2L_2]} D^{(\ell)} \leq \widetilde{O}(\varepsilon^{-2}) + D_w,$$

$$||\boldsymbol{\theta}||_{op} \leq O(1 + \eta) + C_w, \quad \sum_{\ell=1}^{L_1} M^{(\ell)} = d_{\max} + q_{\max}.$$

*where we hide the constants $K, B_x, B_u, B_v, C^4$, satisfies the following*

$$||\widehat{\boldsymbol{w}} - \boldsymbol{w}_{GD}^L||_2 \leq L_f^{-1}(1 + \eta L_f)^L \varepsilon,$$

*where $L_f = \sup_{w \in \mathcal{W}} \left|\left|\nabla^2 \widehat{L}_N(\boldsymbol{w})\right|\right|_2$.*

**Maximum Likelihood Estimation (Gaussian) via Transformers.** The next result shows that MOIRAI transformers are also capable of performing maximum likelihood estimation on any input multi-variate time series. Given a data generated by some $\text{AR}_d(q)$ process with parameter $(\boldsymbol{w}_1, \cdots, \boldsymbol{w}_q) \subset \mathbb{R}^d$: $(\boldsymbol{x}_1, \cdots, \boldsymbol{x}_T) \subset \mathbb{R}^d$, the conditional likelihood $f(\cdot)$ of observing $\boldsymbol{x}_t$ is

$$f(\boldsymbol{x}_t \mid \boldsymbol{x}_{t-1}, \cdots, \boldsymbol{x}_{t-q}) := \frac{1}{\sqrt{2\pi\sigma^2}} \exp\left(\frac{-(\boldsymbol{x}_t - \sum_{i=1}^q \langle \boldsymbol{w}_i, \boldsymbol{x}_{t-i}\rangle)^2}{2\sigma^2}\right).$$

The goal is to estimate the mean vector $(\boldsymbol{w}_1, \cdots, \boldsymbol{w}_q)$ and the variance $\sigma^2$ by minimizing the negative log-likelihood loss. Note that with $n \geq d$, the loss is strongly convex. The optimization over the NLL Loss has two steps: estimating the mean vector: $\widehat{\boldsymbol{w}}$, and then derive the variance $\widehat{\sigma}^2$ with the following closed-form solution:

$$\sigma^2 = \frac{1}{T} \sum_{t=1}^T \left(\boldsymbol{x}_t - \sum_{i=1}^q \langle \widehat{\boldsymbol{w}}_i, \boldsymbol{x}_{t-i}\rangle\right)^2.$$

**Theorem E.5.** *Given a set of input data generated by some $\text{AR}_d(q)$ process: $\boldsymbol{X}, \in \mathbb{R}^{n \times d}, \boldsymbol{Y} \in \mathbb{R}^n$, considering the following negative log-likelihood loss, the goal is to find a set of parameters $\boldsymbol{w} \in \mathbb{R}^d, \sigma^2 \in \mathbb{R}^+$ to minimize the following loss*

$$L_{NLL}(\boldsymbol{w}, \sigma) := \frac{n}{2} \log(2\pi\sigma^2) + \frac{1}{2\sigma^2} \sum_{t=q+1}^T \left(\boldsymbol{x}_t - \sum_{i=1}^q \langle \boldsymbol{w}_i, \boldsymbol{x}_{t-i}\rangle\right)^2$$

*We denote $\boldsymbol{w}^\star, \sigma^\star$ as the ERM satisfying the NLL Loss. There exists a $(L_1 + L_2 + 2)$-layer MOIRAI Transformer such that its first $L_1 + L_2$ layers follow the same requirement in Theorem 3.6, and the last two layers each has two and one heads, it estimates $\boldsymbol{w}, \boldsymbol{\sigma}$ with bounded error:*

$$\|\widehat{\boldsymbol{w}} - \boldsymbol{w}^\star\| \leq \varepsilon,$$

*and the estimated variance is bounded by*

$$\left|\widehat{\sigma}^2 - \sigma^{\star 2}\right| \leq 2EB_x\varepsilon + B_x^2\varepsilon = \widetilde{O}(\varepsilon + \varepsilon^2),$$

*where $E \leq B(1 + B_w)$, and $\widetilde{O}$ hides the values dependent on $B_x, B_w$.*

*Proof of Theorem E.5.*

$$L_{\text{NLL}}(\boldsymbol{w}, \sigma) := \frac{n}{2} \log(2\pi\sigma^2) + \frac{1}{2\sigma^2} \sum_{t=1}^T \left(\boldsymbol{x}_t - \sum_{i=1}^q \langle \boldsymbol{w}_i, \boldsymbol{x}_{t-i}\rangle\right)^2.$$

Following Theorem 3.6, the first $L_1 + L_2$ layers of MOIRAI obtains $\widehat{\boldsymbol{w}}$ such that the $L_1 + L_2 + 1$-th layer takes the following as input

$$\widetilde{\boldsymbol{h}}_i^{(L_1+L_2)} = \left[x_i^1; \boldsymbol{x}_{i-1:i-q}^1; \boldsymbol{x}_{i-1:i-q}^2; \cdots; \boldsymbol{x}_{i-1:i-q}^d; \boldsymbol{w}^\star + \boldsymbol{\varepsilon}; \boldsymbol{0}; 1; t_i\right],$$

where $\boldsymbol{w}^\star + \boldsymbol{\varepsilon} \in \mathbb{R}^{qd}$ is the flatten mean vectors. For the simplicity of notations, for the $i$-th column, we denote $x_i^1$ with $\widetilde{\boldsymbol{y}}_i$, and denote $[\boldsymbol{x}_{i-1:i-q}^1; \boldsymbol{x}_{i-1:i-q}^2; \cdots; \boldsymbol{x}_{i-1:i-q}^d]$ as $\widetilde{\boldsymbol{x}}_i \in \mathbb{R}^{qd}$, as they correspond to the label and feature of our AR model, respectively. $\boldsymbol{\varepsilon} \in \mathbb{R}^{dq}$ satisfies

$$\|\boldsymbol{\varepsilon}\| \leq \varepsilon \cdot (\eta B_x).$$

Now we start to construct the $(L_1 + L_2 + 1)$-th layer. One can then construct

$$\boldsymbol{Q}_1^{L+1}\boldsymbol{h}_i^L = [\boldsymbol{0}; \widetilde{\boldsymbol{x}}_i; \boldsymbol{0}], \quad \boldsymbol{K}_1^{L+1}\boldsymbol{h}_j^L = [\boldsymbol{0}; \widehat{\boldsymbol{w}}; \boldsymbol{0}], \quad \boldsymbol{V}_1^{L+1}\boldsymbol{h}_k^L = [\boldsymbol{0}; 1; \boldsymbol{0}]$$
$$\boldsymbol{Q}_2^{L+1}\boldsymbol{h}_i^L = [\boldsymbol{0}; \widetilde{\boldsymbol{x}}_i; \boldsymbol{0}], \quad \boldsymbol{K}_2^{L+1}\boldsymbol{h}_j^L = [\boldsymbol{0}; -\widehat{\boldsymbol{w}}; \boldsymbol{0}], \quad \boldsymbol{V}_2^{L+1}\boldsymbol{h}_k^L = [\boldsymbol{0}; -1; \boldsymbol{0}].$$

The above construction gives us

$$\boldsymbol{h}_i^{L+1} = \boldsymbol{h}_i^L + \frac{1}{n} \sum_{j=1}^n \sum_{m=1}^2 \sigma\left(\langle \boldsymbol{Q}_m^{L+1}\boldsymbol{h}_{n+1}^L, \boldsymbol{K}_m^{L+1}\boldsymbol{h}_j^L\rangle\right) \boldsymbol{V}_m^{L+1}\boldsymbol{h}_j^L$$
$$= [\widetilde{\boldsymbol{y}}_i; \widetilde{\boldsymbol{x}}_i; \widehat{\boldsymbol{w}}; \boldsymbol{0}; 1; t_i] + (\sigma(\langle \widehat{\boldsymbol{w}}, \boldsymbol{x}_i\rangle) - \sigma(-\langle \widehat{\boldsymbol{w}}, \boldsymbol{x}_i\rangle)) \cdot [\boldsymbol{0}; 1; \boldsymbol{0}]$$
$$= [\widetilde{\boldsymbol{y}}_i; \widetilde{\boldsymbol{x}}_i; \widehat{\boldsymbol{w}}; \langle \widehat{\boldsymbol{w}}, \widetilde{\boldsymbol{x}}_i\rangle, \boldsymbol{0}; 1; t_i].$$

Next, we construct the last layer as

$$\boldsymbol{Q}_1^{L+1}\boldsymbol{h}_i^L = [...; \widetilde{\boldsymbol{y}}_i - \langle\widehat{\boldsymbol{w}}, \widetilde{\boldsymbol{x}}_i\rangle; ...], \quad \boldsymbol{K}_1^{L+1}\boldsymbol{h}_j^L = [...; \widetilde{\boldsymbol{y}}_j - \langle\widehat{\boldsymbol{w}}, \widetilde{\boldsymbol{x}}_j\rangle; ...], \quad \boldsymbol{V}_1^{L+1}\boldsymbol{h}_k^L = [\boldsymbol{0}; 1; \boldsymbol{0}]$$

Finally, the result becomes

$$\boldsymbol{h}_i = \frac{1}{n}[...; \sum_{\mu=1}^n (\boldsymbol{y}_\mu - \langle\boldsymbol{x}_\mu, \widehat{\boldsymbol{w}}\rangle)^2; ...] = [...; \widehat{\sigma^2}; ...].$$

Thus, we complete the proof. □

E.3    PROOF OF THE LIPSCHITZNESS OF ANY-VARIATE TRANSFORMERS

We first show the Lipschitzness of each component in an Any-Variate Transformer. For any $p \in [1, \infty]$, let $||\boldsymbol{H}||_{2,p} := (\sum_{i=1}^{N} ||\boldsymbol{h}_i||_2^p)^{1/p}$ denote the column-wise $(2, p)$-norm of $\boldsymbol{H}$. For any radius $\mathtt{R} > 0$, we denote $\mathcal{H}_{\mathtt{R}} := \{\boldsymbol{H} : ||\boldsymbol{H}||_{2,\infty} \leq \mathtt{R}\}$ be the ball of radius $\mathtt{R}$ under norm $||\cdot||_{2,\infty}$.

**Lemma E.6.** *For a single Any-Variate attention layer,* $\boldsymbol{\theta}_1 = \{(\boldsymbol{V}_m, \boldsymbol{Q}_m, \boldsymbol{K}_m, u_m^1, u_m^2)\}_{m \in [M]}$*, we introduce its norm*

$$\|\boldsymbol{\theta}_1\| := \max_{m \in [M]} \{||\boldsymbol{Q}_m||_{op}, ||\boldsymbol{K}_m||_{op}, |u_m^1|, |u_m^2|\} + \sum_{m=1}^{M} ||\boldsymbol{V}_m||_{op}$$

*For any fixed hidden dimension* $D'$*, we consider*

$$\Theta_{1,B} := \{\boldsymbol{\theta}_1 : |||\boldsymbol{\theta}_1||| \leq B\}.$$

*Then for* $\boldsymbol{H} \in \mathcal{H}_{\mathtt{R}}$*,* $\boldsymbol{\theta}_1 \in \Theta_{1,B}$*, the function* $(\boldsymbol{\theta}_1, \boldsymbol{H}) \mapsto Attn_{\boldsymbol{\theta}_1}$ *is* $(1 + \iota)$*-Lipschitz w.r.t.* $\boldsymbol{\theta}_1$*, where* $\iota = \max\{B^2\mathtt{R}^2 + T + (T-1)d, B(T-1)d\}$*, and* $(1 + B^3\mathtt{R}^2)$*-Lipschitz w.r.t.* $\boldsymbol{H}$*.*

*Proof.* Given some $\epsilon > 0$, some set $X$ and a function class $\mathcal{F}$. If $\mathcal{F}$ is $L$-Lipschitzness, i.e.,

$$||f(x_1) - f(x_2)|| \leq L ||x_1 - x_2||, \quad \text{for all } f \in \mathcal{F}.$$

Then, the following holds

$$N(\epsilon, \mathcal{F}, ||\cdot||) \leq N(\epsilon/L, X, ||\cdot||).$$

Define

$$\Theta_{\text{attn},B} := \{\theta_{\text{attn}} : |||\theta_{\text{attn}}||| \leq B\}.$$

The output of the Any-Variate Attention $[\widetilde{h}_i]$ is given by

$$\widetilde{h}_i = h_i + \sum_{m=1}^{M} \frac{1}{N} \sum_{j=1}^{N} \sigma\left(\langle \mathbf{Q}_m h_i, \mathbf{K}_m h_j \rangle \cdot \mathbf{V}_m h_j + u_m^1 \star \mathbf{U} + u_m^2 \star \bar{\mathbf{U}}\right).$$

We also define $\theta'_{\text{attn}} = \{\mathbf{V}'_m, \mathbf{Q}'_m, \mathbf{K}'_m, u_m^{1\prime}, u_m^{2\prime}\}_{m \in [M]}$. $\widetilde{h}'_i$ as

$$\widetilde{h}'_i = h_i + \sum_{m=1}^{M} \frac{1}{N} \sum_{j=1}^{N} \sigma\left(\langle \mathbf{Q}'_m h_i, \mathbf{K}'_m h_j \rangle \cdot \mathbf{V}'_m h_j + u_m^{1\prime} \star \mathbf{U} + u_m^{2\prime} \star \bar{\mathbf{U}}\right).$$

Now we bound $\left|\left|Attn_{\boldsymbol{\theta}_1}(\boldsymbol{H}) - Attn_{\boldsymbol{\theta}'_1}(\boldsymbol{H})\right|\right|_{2,\infty} = \max_i \left|\left|\widetilde{h}_i - \widetilde{h}'_i\right|\right|_2$ as follows

$$\left|\left|\widetilde{\boldsymbol{h}}_i - \widetilde{\boldsymbol{h}}'_i\right|\right|_2 = \left|\left| \sum_{m=1}^{M} \frac{1}{N} \left[ \sum_{j=1}^{N} \sigma\left(\langle \mathbf{Q}_m h_i, \mathbf{K}_m h_j \rangle + u_m^1 \star \mathbf{U} + u_m^2 \star \bar{\mathbf{U}}\right) \mathbf{V}_m h_j - \right. \right. \right.$$

$$\left. \left. \left. \sum_{j=1}^{N} \sigma\left(\langle \mathbf{Q}'_m h_i, \mathbf{K}'_m h_j \rangle + u_m^{1\prime} \star \mathbf{U} + u_m^{2\prime} \star \bar{\mathbf{U}}\right) \mathbf{V}'_m h_j \right] \right|\right|_2$$

$$\leq \sum_{m=1}^{M} \frac{1}{N} \sum_{j=1}^{N} \left|\left| \sigma\left(\langle \mathbf{Q}_m h_i, \mathbf{K}_m h_j \rangle + u_m^1 \star \mathbf{U} + u_m^2 \star \bar{\mathbf{U}}\right) \mathbf{V}_m h_j \right.\right.$$

$$\left.\left. - \sigma\left(\langle \mathbf{Q}'_m h_i, \mathbf{K}'_m h_j \rangle + u_m^{1\prime} \star \mathbf{U} + u_m^{2\prime} \star \bar{\mathbf{U}}\right) \mathbf{V}'_m h_j \right|\right|_2$$

$$\leq \sum_{m=1}^{M} \frac{1}{N} \sum_{j=1}^{N} ||h_j||_2 \left|\left| \sigma\left(\langle \mathbf{Q}_m h_i, \mathbf{K}_m h_j \rangle + u_m^1 \star \mathbf{U} + u_m^2 \star \bar{\mathbf{U}}\right) \mathbf{V}_m \right.\right.$$

$$\left.\left. - \sigma\left(\langle \mathbf{Q}'_m h_i, \mathbf{K}'_m h_j \rangle + u_m^{1\prime} \star \mathbf{U} + u_m^{2\prime} \star \bar{\mathbf{U}}\right) \mathbf{V}'_m \right|\right|_{op}.$$

Let

$$A = \langle \mathbf{Q}_m h_i, \mathbf{K}_m h_j \rangle + u_m^1 \star \mathbf{U} + u_m^2 \star \bar{\mathbf{U}}$$
$$B = \langle \mathbf{Q}'_m h_i, \mathbf{K}'_m h_j \rangle + u_m^{1'} \star \mathbf{U} + u_m^{2'} \star \bar{\mathbf{U}}.$$

By triangle inequality, we have

$$||\sigma(A)V_m - \sigma(B)V'_m|| \leq ||\sigma(A)||_{\mathrm{op}} ||\boldsymbol{V}_m - \boldsymbol{V}'_m||_{\mathrm{op}} + ||\sigma(A) - \sigma(B)||_{\mathrm{op}} ||\boldsymbol{V}'_m||_{\mathrm{op}}.$$

Note that $\sigma(\cdot)$ is 1-Lipschitz, we get

$$||\sigma(A) - \sigma(B)||_{\mathrm{op}} \leq ||A - B||_{\mathrm{op}}$$
$$= \left|\left|\langle \mathbf{Q}_m h_i, \mathbf{K}_m h_j \rangle - \langle \mathbf{Q}'_m h_i, \mathbf{K}'_m h_j \rangle + (u_m^1 - u_m^{1'})\boldsymbol{U} + (u_m^2 - u_m^{2'})\widehat{\boldsymbol{U}}\right|\right|_{\mathrm{op}}$$
$$\leq \left|\left|\left|\langle \mathbf{Q}_m h_i, \mathbf{K}_m h_j \rangle - \langle \mathbf{Q}'_m h_i, \mathbf{K}'_m h_j \rangle\right| + \left|\left|\left|(u_m^1 - u_m^{1'}) \star \boldsymbol{U}\right|\right|\right| + \left|\left|\left|(u_m^2 - u_m^{2'}) \star \bar{\boldsymbol{U}}\right|\right|\right|\right|.$$

For the first term in the last inequality, we have

$$\langle \mathbf{Q}_m h_i, \mathbf{K}_m h_j \rangle - \langle \mathbf{Q}'_m h_i, \mathbf{K}'_m h_j \rangle \leq ||\boldsymbol{Q}_m - \boldsymbol{Q}'_m|| \, ||h_i|| \, ||h_j|| \, ||\boldsymbol{K}_m|| + ||\boldsymbol{K}_m - \boldsymbol{K}'_m|| \, ||h_i|| \, ||h_j|| \, ||\boldsymbol{Q}_m||$$
$$= \mathtt{R}^2 B \left(||\boldsymbol{Q}_m - \boldsymbol{Q}'_m|| + ||\boldsymbol{K}_m - \boldsymbol{K}'_m||\right).$$

Further, we have

$$\left|\left|(u_m^1 - u_m^{1'}) \star \boldsymbol{U}\right|\right| \leq |u_m^1 - u_m^{1'}| \, ||\boldsymbol{U}|| \leq T|u_m^1 - u_m^{1'}|,$$

where $T$ is the length of each variate (lookback window size).

$$\left|\left|(u_m^2 - u_m^{2'}) \star \bar{\boldsymbol{U}}\right|\right| \leq |u_m^2 - u_m^{2'}| \, ||\bar{\boldsymbol{U}}|| \leq (T-1)d|u_m^2 - u_m^{2'}|,$$

where $d$ is the number of variates.

Thus, we have

$$||\sigma(A) - \sigma(B)||_{\mathrm{op}} ||\boldsymbol{V}'_m||_{\mathrm{op}} \leq B\left(\mathtt{R}^2 B\left(||\boldsymbol{Q}_m - \boldsymbol{Q}'_m|| + ||\boldsymbol{K}_m - \boldsymbol{K}'_m||\right) + T\left(|u_m^1 - u_m^{1'}|\right) + (T-1)d\left(|u_m^2 - u_m^{2'}|\right)\right)$$
$$\leq B \cdot \max\{\mathtt{R}^2 B, (T-1)d\} \cdot \left(||\boldsymbol{Q}_m - \boldsymbol{Q}'_m|| + ||\boldsymbol{K}_m - \boldsymbol{K}'_m|| + |u_m^1 - u_m^{1'}| + |u_m^2 - u_m^{2'}|\right).$$

Next, we bound

$$||\sigma(A)||_{\mathrm{op}} \leq ||A||_{\mathrm{op}} \leq B^2 \mathtt{R}^2 + (T + (T-1)d),$$

due to the fact that

$$||A|| \leq ||\boldsymbol{Q}_m h_i|| \, ||\boldsymbol{K}_m h_j|| \, ||u_m^1 \boldsymbol{U}|| \, ||u_m^2 \bar{\boldsymbol{U}}||.$$

Overall, the Any-Variate Attention is $\max\{B^2 \mathtt{R}^2 + T + (T-1)d, B(T-1)d\}$-Lipshcitz in $\boldsymbol{\theta}_1$.  $\square$

*Proof.* We start by considering $\boldsymbol{H}' = [\boldsymbol{h}'_i]$ and

$$\widetilde{\boldsymbol{h}}'_i = \boldsymbol{h}'_i + \sum_{m=1}^{M} \frac{1}{N} \sum_{j=1}^{N} \sigma\left(\langle \mathbf{Q}_m \boldsymbol{h}'_i, \mathbf{K}_m \boldsymbol{h}'_j \rangle + u_m^1 \cdot \mathbf{U} + u_m^2 \cdot \bar{\mathbf{U}}\right) \cdot \mathbf{V}_m \boldsymbol{h}'_j.$$

We then bound

$$\left\|\left(\widetilde{\boldsymbol{h}}_i' - \boldsymbol{h}_i'\right) - \left(\widetilde{\boldsymbol{h}}_i - \boldsymbol{h}_i\right)\right\|_2$$

$$= \left\|\sum_{m=1}^{M} \frac{1}{N} \sum_{j=1}^{N} \left[\sigma\left(\langle \mathbf{Q}_m \boldsymbol{h}_i', \mathbf{K}_m \boldsymbol{h}_j'\rangle + u_m^1 \cdot \mathbf{U} + u_m^2 \cdot \bar{\mathbf{U}}\right) \mathbf{V}_m \boldsymbol{h}_j - \left(\langle \mathbf{Q}_m \boldsymbol{h}_i', \mathbf{K}_m \boldsymbol{h}_j'\rangle + u_m^1 \cdot \mathbf{U} + u_m^2 \cdot \bar{\mathbf{U}}\right) \mathbf{V}_m \boldsymbol{h}_j'\right]\right\|_2$$

$$\leq \sum_{m=1}^{M} \frac{1}{N} \sum_{j=1}^{N} \|\mathbf{V}_m\|_{\mathrm{op}} \left\|\sigma\left(\langle \mathbf{Q}_m \boldsymbol{h}_i', \mathbf{K}_m \boldsymbol{h}_j'\rangle + u_m^1 \cdot \mathbf{U} + u_m^2 \cdot \bar{\mathbf{U}}\right) \boldsymbol{h}_j - \left(\langle \mathbf{Q}_m \boldsymbol{h}_i', \mathbf{K}_m \boldsymbol{h}_j'\rangle + u_m^1 \cdot \mathbf{U} + u_m^2 \cdot \bar{\mathbf{U}}\right) \boldsymbol{h}_j'\right\|_2$$

$$\leq \sum_{m=1}^{M} \frac{1}{N} \sum_{j=1}^{N} \|\mathbf{V}_m\|_{\mathrm{op}} \left\{|\sigma\left(\langle \mathbf{Q}_m \boldsymbol{h}_i, \mathbf{K}_m \boldsymbol{h}_j\rangle + u_m^1 \mathbf{U} + u_m^2 \bar{\mathbf{U}}\right)| \cdot \left\|\boldsymbol{h}_j - \boldsymbol{h}_j'\right\|_2 \right.$$

$$+ |\sigma\left(\langle \mathbf{Q}_m \boldsymbol{h}_i, \mathbf{K}_m \boldsymbol{h}_j\rangle + u_m^1 \mathbf{U} + u_m^2 \bar{\mathbf{U}}\right) - \sigma\left(\langle \mathbf{Q}_m \boldsymbol{h}_i', \mathbf{K}_m \boldsymbol{h}_j\rangle + u_m^1 \mathbf{U} + u_m^2 \bar{\mathbf{U}}\right)| \cdot \left\|\boldsymbol{h}_j'\right\|_2$$

$$\left. + |\sigma\left(\langle \mathbf{Q}_m \boldsymbol{h}_i, \mathbf{K}_m \boldsymbol{h}_j\rangle + u_m^1 \mathbf{U} + u_m^2 \bar{\mathbf{U}}\right) - \sigma\left(\langle \mathbf{Q}_m \boldsymbol{h}_i', \mathbf{K}_m \boldsymbol{h}_j'\rangle + u_m^1 \mathbf{U} + u_m^2 \bar{\mathbf{U}}\right)| \left\|\boldsymbol{h}_j'\right\|_2 \right\}$$

$$\leq \sum_{m=1}^{M} \frac{1}{N} \sum_{j=1}^{N} \|\mathbf{V}_m\|_{\mathrm{op}} \cdot 3 \|\mathbf{Q}_m\|_{\mathrm{op}} \|\mathbf{K}_m\|_{\mathrm{op}} \mathtt{R}^2 \left\|\boldsymbol{h}_j - \boldsymbol{h}_j'\right\|_2$$

$$\leq B^3 \mathtt{R}^2 \left\|\boldsymbol{H} - \boldsymbol{H}'\right\|_{2,\infty}.$$

Where the third inequality comes from the fact that ReLU is 1-Lipschitzness, and the fourth and fifth inequality comes from the AM-GM inequality. For more details, refer (5, Section J.2) $\square$

**Corollary E.7** (Lipschitz Constant of Single Layer Moirai Transformer). *For a fixed number of heads $M$ and hidden dimension $D'$, we consider*

$$\Theta_{TF,1,B} = \{\boldsymbol{\theta} = (\boldsymbol{\theta}_1, \boldsymbol{\theta}_2)\} : M \text{ heads, hidden dimension } D', \|\boldsymbol{\theta}\|_{op} \leq B.$$

*Then for the function $TF^{\mathtt{R}}$ given by*

$$TF^{\mathtt{R}} : (\boldsymbol{\theta}, \boldsymbol{H}) \mapsto \mathtt{clip}_{\mathtt{R}}(MLP_{\boldsymbol{\theta}_2}(Attn_{\boldsymbol{\theta}_1}(\boldsymbol{H}))), \quad \boldsymbol{\theta} \in \Theta_{TF,1,B}, \boldsymbol{H} \in \mathcal{H}_{\mathtt{R}}.$$

*$TF^{\mathtt{R}}$ is $B_\Theta$-Lipschitz w.r.t. $\boldsymbol{\theta}$ and $B_H$-Lipschitz w.r.t. $\boldsymbol{H}$, where $B_\Theta = (1+B^2)(1+\iota) + B\mathtt{R}(1+B^3\mathtt{R}^2)$ and $B_H = (1+B^2)(1+B^3\mathtt{R}^2)$.*

**Proposition E.8** (Lipschitz Constant of Moirai Transformer). *For a fixed number of heads $M$ and hidden dimension $D'$, we consider*

$$\Theta_{TF,L,B} = \{\boldsymbol{\theta} = (\boldsymbol{\theta}_1^{(1:L)}, \boldsymbol{\theta}_2^{(1:L)})\} : M^{(\ell)} = M, D^{(\ell)} = D', \|\boldsymbol{\theta}\|_{op} \leq B.$$

*Then for the function $TF^{\mathtt{R}}$ is $(LB_H^{L-1}B_\Theta)$-Lipschitz in $\boldsymbol{\theta} \in \Theta_{TF,L,B}$ for any fixed $\boldsymbol{H}$.*

*Proof.* For any $\boldsymbol{\theta} = (\boldsymbol{\theta}_1, \boldsymbol{\theta}_2)$, $\boldsymbol{H} \in \mathcal{H}_{\mathtt{R}}$, and $\theta' = (\theta_1', \theta_2')$, we have

$$\|TF_{\boldsymbol{\theta}}(\boldsymbol{H}) - TF_{\theta'}(\boldsymbol{H})\|_{2,\infty} \leq \left\|MLP_{\boldsymbol{\theta}_2}(Attn_{\boldsymbol{\theta}_1}(\boldsymbol{H})) - MLP_{\boldsymbol{\theta}_2}(Attn_{\theta_1'}(\boldsymbol{H}))\right\|_{2,\infty} +$$

$$\left\|MLP_{\boldsymbol{\theta}_2}(Attn_{\theta_1'}(\boldsymbol{H})) - MLP_{\theta_2'}(Attn_{\theta_1'}(\boldsymbol{H}))\right\|_{2,\infty}$$

$$\leq (1 + B^2) \left\|Attn_{\theta_1'}(\boldsymbol{H}) - Attn_{\theta_1'}(\boldsymbol{H})\right\|_{2,\infty} + B\bar{\mathtt{R}} \|\boldsymbol{\theta}_2 - \theta_2'\|_{op}$$

$$\leq (1 + B^2)(1 + \iota) \|\boldsymbol{\theta}_1 - \theta_1'\|_{op} + B\bar{\mathtt{R}} \|\boldsymbol{\theta}_2 - \theta_2'\|_{op} \leq B_\Theta \|\boldsymbol{\theta} - \theta'\|_{op},$$

where $\bar{\mathtt{R}} = \mathtt{R} + B^3\mathtt{R}^3$, $\iota = \max\{B^2\mathtt{R}^2 + T + (T-1)d, B(T-1)d\}$. The second inequality comes from the fact $\|Attn_{\boldsymbol{\theta}}(\boldsymbol{H})\| \leq \mathtt{R} + B^3\mathtt{R}^3$.

Further, for $\boldsymbol{H}' \in \mathcal{H}_{\mathtt{R}}$, we have

$$\|TF_{\boldsymbol{\theta}}(\boldsymbol{H}) - TF_{\boldsymbol{\theta}}(\boldsymbol{H}')\|_{2,\infty} \leq (1 + B^2) \|Attn_{\boldsymbol{\theta}_1}(\boldsymbol{H}) - Attn_{\boldsymbol{\theta}_1}(\boldsymbol{H}')\|$$

$$\leq (1 + B^2)(1 + B^3\mathtt{R}^2) \|\boldsymbol{H} - \boldsymbol{H}'\|_{2,\infty}.$$

For the multi-layer case, one can simply follow (5, Proposition J.1) to conclude the proof.

$\square$

### E.4 PROOF OF THEOREM 4.4

We first describe steps of our proving technique .

    (i) We write out the pretraining data generation and objective.

    (ii) We show that the objective of pretraining on $n$ time series satisfying Dobrushins condition with length $T$, is equivalent to pretraining on a single time series with length $nT$ under Dobrushin's condition with the same coefficient.

    (iii) We bound the complexity of transformers, and apply learning bounds from (7).

Let $\pi$ be a meta distribution, and each distribution drawn from $\mathrm{P}^{(T)} \sim \pi$ satisfies the Dobrushin's condition with max coefficient as $\alpha$. We then define the single-path average loss as

$$Y_{\theta, \mathrm{P}^{(T)}} := \frac{1}{T} \sum_{t=1}^{T} \ell(\theta, \boldsymbol{z}_t) - \mathbb{E}_{\boldsymbol{z} \sim \mathrm{P}^{(T)}} \left[ \ell(\theta, \boldsymbol{z}) \right].$$

Now, we assume our pretraining data is generated by the following

    1. Sample $n$ distributions from $\pi$ i.i.d. to get $\mathrm{P}_j^{(T)}$, for $j = 1, \cdots, n$

    2. For each distribution $\mathrm{P}_j^{(T)}$, we sample $(\boldsymbol{z}_{j,1}, \cdots, \boldsymbol{z}_{j,T})$

**Assumption E.9.** *We assume that for each $j \in [n]$, $(z_{j,t})$ has marginals equal to some distribution $D$ for $t = 1, \cdots, T$.*

We first present several lemma and theorems that will be used later.

**Lemma E.10** ((38, Example 5.8)). *Given any well-defined norm $||\cdot||'$. Let $\mathbb{B}$ be the $\mathbb{R}^d$ unit-ball in $||\cdot||'$, i.e. $\mathbb{B} = \{\theta \in \mathbb{R}^d \mid ||\theta||' \leq 1\}$, we have*

$$\log N(\delta, \mathbb{B}, ||\cdot||') \leq d \log \left( 1 + \frac{2}{\delta} \right).$$

**Theorem E.11** ((7, Theorem 5.3)). *Given a function class $\mathscr{F}$, such that $|f| \leq B$, for all $f \in \mathscr{F}$. Let $\mathrm{P}^{(T)}$ be a distribution over some domain $Z^{(T)}$, assuming Assumption E.9 holds and $\alpha_{\log}(\mathrm{P}^{(T)}) < 1/2$. Then for all $t > 0$,*

$$P_{\boldsymbol{z} \sim \mathrm{P}^{(T)}} \left( \sup_{f \in \mathscr{F}} \left| \frac{1}{T} \sum_{i=1}^{T} f(z_i) - \mathbb{E}_z[f(z)] \right| > C \left( \mathfrak{G}_{\mathrm{P}^{(T)}}(\mathscr{F}) + \frac{Bt}{\sqrt{T}} \right) \right) \leq e^{-t^2/2},$$

*for some universal constant whenever $1/2 - \alpha_{\log}(\mathrm{P}^{(T)})$ is bounded away from zero.*

The following theorem is from (7; 17).

**Theorem E.12.** *Let $\mathrm{P}_{\boldsymbol{z}}^{(T)}$ be a distribution satisfying the Dobrushin's condition with coefficient $\alpha(\mathrm{P}_{\boldsymbol{z}}^{(T)})$. Let $(\boldsymbol{z}_1, \cdots, \boldsymbol{z}_T) \sim \mathrm{P}^{(T)}$, and let $f : \boldsymbol{Z}^{(T)} \to \mathbb{R}$ be a real-valued function with the following bounded difference property, with parameters $\lambda_1, \cdots, \lambda_T \geq 0$:*

$$|f(\boldsymbol{z}) - f(\boldsymbol{z}')| \leq \sum_{t=1}^{T} \mathbb{1}_{\boldsymbol{z}_t \neq \boldsymbol{z}'_t} \lambda_t.$$

*Then for all $t > 0$,*

$$P\left( |f(\boldsymbol{z}) - \mathbb{E}[f(\boldsymbol{z})] \geq t \right) \leq 2 \exp \left( -\frac{(1 - \alpha)t^2}{2 \sum_t \lambda_t^2} \right)$$

The following corollary directly follows from the above result

**Corollary E.13.** *Following Theorem E.12, let*

$$\ell(\boldsymbol{z}) := \frac{1}{T} \sum_{t=1}^{T} \ell(\boldsymbol{z}_t),$$

*where $0 \leq \ell(\boldsymbol{z}_t) \leq B$ for all $t = 1, \cdots, T$ and all $\boldsymbol{z} \sim \mathtt{P}_{\boldsymbol{z}}$. Then the variance of $\ell(\cdot)$ is bounded by*

$$|\ell(\boldsymbol{z}) - \ell(\boldsymbol{z}')| \leq B.$$

*Then, the following holds*

$$P\left(|\ell(\boldsymbol{z}) - \mathtt{E}[\ell(\boldsymbol{z})]| \geq t\right) \leq 2 \exp\left(\frac{-(1-\alpha)t^2}{2\sum_t B^2}\right).$$

**Direct Application of Theorem E.11.** By Theorem E.11, if Assumption E.9 holds, with probability over $1 - e^{-t^2/2}$, for any $\theta \in \Theta$, $\alpha_{\log}(\mathtt{P}_j^{(T)}) < 1/2$ we have

$$\sup_{\theta \in \Theta} \left|Y_{\theta, \mathtt{P}^{(T)}}\right| \leq C \left[\mathfrak{G}_{\mathtt{P}_j^{(T)}}(\ell(\Theta)) + \frac{2tB_x^2}{\sqrt{T}}\right],$$

where $\ell(\Theta)$ denotes the function class of $\ell(\theta, )$, for all $\theta \in \Theta$, and $C > 0$ is an universal constant. Note that the above bound presents the naive learning bound for learning a single time series, which is a direct result from (7).

Here we show properties of a time series generated by concatenating $n$ series under Dobrushin's condition with bounded coefficient.

**Lemma E.14** (Dobrushin submultiplicativity)**.** *Let $P, Q$ be Markov kernels on a measurable space $(\mathsf{X}, \mathscr{F})$ with Dobrushin coefficients*

$$\alpha(P) := \sup_{x,x'} \|P(x, \cdot) - P(x', \cdot)\|_{\mathrm{TV}}, \qquad \alpha(Q) := \sup_{x,x'} \|Q(x, \cdot) - Q(x', \cdot)\|_{\mathrm{TV}}.$$

*Then $\alpha(PQ) \leq \alpha(P)\,\alpha(Q)$.*

*Proof.* For any probability measures $\mu, \nu$ on $\mathsf{X}$,

$$\|\mu PQ - \nu PQ\|_{\mathrm{TV}} = \|(\mu P - \nu P)Q\|_{\mathrm{TV}} \leq \alpha(Q) \|\mu P - \nu P\|_{\mathrm{TV}} \leq \alpha(Q)\,\alpha(P) \|\mu - \nu\|_{\mathrm{TV}}.$$

Taking $\sup_{\mu,\nu}$ over Dirac measures yields $\alpha(PQ) \leq \alpha(P)\alpha(Q)$. $\qquad\square$

**Lemma E.15** (Concatenation preserves Dobrushin)**.** *Let $\{P_t\}_{t=1}^T$ be kernels with $\alpha(P_t) \leq \kappa_t < 1$. For the $T$-step kernel $P_{1:T} := P_1 P_2 \cdots P_T$,*

$$\alpha(P_{1:T}) \leq \prod_{t=1}^T \kappa_t.$$

*In particular, if $\kappa_t \leq \kappa < 1$ for all $t$, then $\alpha(P_{1:T}) \leq \kappa^T$.*

*Proof.* Apply the previous lemma repeatedly:

$$\alpha(P_{1:T}) \leq \prod_{t=1}^T \alpha(P_t) \leq \prod_{t=1}^T \kappa_t.$$

$$\square$$

**Corollary E.16** (Concatenating $n$ Dobrushin blocks)**.** *Let $\{P_t^{(j)}\}_{t=1}^T$ for $j = 1, \ldots, n$ be kernels for $n$ blocks, and suppose $\alpha\left(P_t^{(j)}\right) \leq \kappa_t^{(j)} < 1$. For the concatenated $nT$-step chain*

$$\underbrace{P_{1:T}^{(1)}}_{\text{block 1}} \underbrace{P_{1:T}^{(2)}}_{\text{block 2}} \cdots \underbrace{P_{1:T}^{(n)}}_{\text{block n}},$$

*its per-step coefficients are bounded by $\kappa_* := \max_{j,t} \kappa_t^{(j)} < 1$, hence the entire $nT$-step kernel has*

$$\alpha\left(\prod_{j=1}^n \prod_{t=1}^T P_t^{(j)}\right) \leq \prod_{j=1}^n \prod_{t=1}^T \kappa_t^{(j)} \leq \kappa_*^{\,nT}.$$

*If all blocks share a common bound $\kappa < 1$, then the concatenated process is Dobrushin with the same per-step bound $\kappa$ and $nT$-step contraction $\leq \kappa^{nT}$.*

*Proof.* A direct result after applying Lemma E.15. □

**Remark E.17.** *A "reset" step (next state independent of current) has $\alpha = 0$ and only improves the bounds. The above results tell us that by concatenating $n$ independent length $T$ Dobrushin time series, we end up having a length $nT$ Dobrushin time series.*

**Concatenation with resets.** For block $j \in [n]$, let $(P_t^{(j)})_{t=1}^T$ be the Markov kernels and let $\mu^{(j)}$ be its prescribed initial law. Define the *reset* kernel $R^{(j)}(x, \cdot) \equiv \mu^{(j)}(\cdot)$. The concatenated $nT$-step chain has kernels

$$\underbrace{P_1^{(1)}, \ldots, P_T^{(1)}}_{\text{block 1}}, \ R^{(1)}, \ \underbrace{P_1^{(2)}, \ldots, P_T^{(2)}}_{\text{block 2}}, \ R^{(2)}, \ \ldots, \ \underbrace{P_1^{(n)}, \ldots, P_T^{(n)}}_{\text{block } n}.$$

**Lemma E.18.** *For any probability measure $\mu$, the reset kernel $R(x, \cdot) \equiv \mu(\cdot)$ satisfies $\alpha(R) = 0$. Consequently, if $\alpha(P_t^{(j)}) \leq \kappa_t^{(j)} < 1$ for all blocks and steps, then the concatenated chain with resets has*

$$\alpha\left( \prod_{j=1}^n \left( P_1^{(j)} \cdots P_T^{(j)} R^{(j)} \right) \right) \ \leq \ \prod_{j=1}^n \prod_{t=1}^T \kappa_t^{(j)}.$$

**Lemma E.19** (Reduction to a single chain of length $nT$). *Let $\widetilde{\mathsf{P}}^{(nT)}$ be the concatenation with resets. Then: (a) $\alpha_{\log}(\widetilde{P}_s) \leq \bar{\alpha} < \frac{1}{2}$ for all $s \leq nT$; (b) the marginal at every time under $\widetilde{\mathsf{P}}^{(nT)}$ equals $D$; (c) for every $\theta$,*

$$\mathbb{E}_{\widetilde{\mathsf{P}}^{(nT)}}\left[ \frac{1}{nT} \sum_{t=1}^{nT} \ell(\theta, \boldsymbol{z}_t) \right] = \frac{1}{n} \sum_{j=1}^n \mathbb{E}_{\mathsf{P}_j^{(T)}}\left[ \frac{1}{T} \sum_{t=1}^T \ell(\theta, \boldsymbol{z}_{j,t}) \right].$$

*Proof.* (a) Direct result from the submultiplicative property of Dobrushin series; resets satisfy $\alpha_{\log}(R^{(j)}) = 0$, hence $\sup_s \alpha_{\log}(\widetilde{P}_s) \leq \bar{\alpha}$. (b) By $DP_t^{(j)} = D$ and $R^{(j)}$ having output $D$, induction over time yields marginal $D$ at all positions. (c) Linearity and the blocking $t = (j-1)T + s$. □

**Lemma E.20.** *Let*

$$X_\theta := \frac{1}{n} \sum_{j=1}^n \left( \frac{1}{T} \sum_{t=1}^T \ell(\theta, \boldsymbol{z}_{j,t}) - \mathbb{E}_{\mathsf{P}_j^{(T)}}\left[ \frac{1}{T} \sum_{t=1}^T \ell(\theta, \boldsymbol{z}_{j,t}) \right] \right)$$

*and*

$$\widetilde{Y}_\theta := \frac{1}{nT} \sum_{t=1}^{nT} \ell(\theta, \boldsymbol{z}_t) - \mathbb{E}_{\widetilde{\mathsf{P}}^{(nT)}}\left[ \frac{1}{nT} \sum_{t=1}^{nT} \ell(\theta, \boldsymbol{z}_t) \right].$$

*Then $\{X_\theta\}_\theta$ and $\{\widetilde{Y}_\theta\}_\theta$ have the same expectation, i.e.*

$$\mathbb{E}_{\widetilde{\mathsf{P}}^{(nT)}}\left[ \frac{1}{nT} \sum_{t=1}^{nT} g(z_t) \right] = \frac{1}{n} \sum_{j=1}^n \mathbb{E}_{\mathsf{P}_j^{(T)}}\left[ \frac{1}{T} \sum_{t=1}^T g(z_{j,t}) \right].$$

*Proof of Lemma E.20.* For all $x, x'$, $R(x, \cdot) = R(x', \cdot) = \mu$, hence $\|R(x, \cdot) - R(x', \cdot)\|_{\mathrm{TV}} = 0$, so $\alpha(R) = 0$. Submultiplicativity gives $\alpha(KL) \leq \alpha(K)\alpha(L)$ and $\alpha(R^{(j)}) = 0$. Use linearity for the expectation identity. □

**Remark E.21** (Equality in law of objectives). *Under Lemma E.20, for each fixed $\theta$, $X_\theta$ and $\widetilde{Y}_\theta$ have the same mean (0) and the same Lipschitz modulus in the sample coordinates. In particular, it suffices to bound $\sup_\theta |\widetilde{Y}_\theta|$.*

**Lemma E.22** (Increment control in $\theta$). *For any $(z_1, \ldots, z_{nT})$,*

$$\left| \frac{1}{nT} \sum_{t=1}^{nT} f_\theta(z_t) - \frac{1}{nT} \sum_{t=1}^{nT} f_{\theta'}(z_t) \right| \leq L_\Theta \, \rho(\theta, \theta').$$

*Hence $\widetilde{Y}_\theta - \widetilde{Y}_{\theta'}$ is subgaussian with variance proxy $\frac{L_\Theta^2 \, \rho(\theta, \theta')^2}{(1 - \kappa_*) \, nT}$.*

*Proof.* Direct application of He et al. (15, Theorem 2.3). □

**Lemma E.23** (Suprema coincide under concatenation). *Let $X_\theta$ and $\widetilde{Y}_\theta$ be*

$$X_\theta = \frac{1}{n} \sum_{j=1}^{n} \left( \frac{1}{T} \sum_{t=1}^{T} \ell(\theta, z_{j,t}) - \mathbb{E}_{\mathsf{P}_j^{(T)}} \left[ \frac{1}{T} \sum_{t=1}^{T} \ell(\theta, z_{j,t}) \right] \right),$$

$$\widetilde{Y}_\theta = \frac{1}{nT} \sum_{t=1}^{nT} \ell(\theta, z_t) - \mathbb{E}_{\widetilde{\mathsf{P}}^{(nT)}} \left[ \frac{1}{nT} \sum_{t=1}^{nT} \ell(\theta, z_t) \right],$$

*where $(z_1, \ldots, z_{nT})$ is the concatenation (with resets) of the blocks $\{(z_{j,1}, \ldots, z_{j,T})\}_{j=1}^{n}$. Then for every realization of the data and for every $\theta$,*

$$X_\theta = \widetilde{Y}_\theta, \quad \text{hence} \quad \sup_{\theta \in \Theta} |X_\theta| = \sup_{\theta \in \Theta} |\widetilde{Y}_\theta|.$$

*Proof.* Write

$$X_\theta = \frac{1}{nT} \sum_{j=1}^{n} \sum_{t=1}^{T} \ell(\theta, z_{j,t}) - \frac{1}{n} \sum_{j=1}^{n} \mathbb{E}_{\mathsf{P}_j^{(T)}} \left[ \frac{1}{T} \sum_{t=1}^{T} \ell(\theta, z_{j,t}) \right].$$

By construction $(z_1, \ldots, z_{nT})$ is just the stacked sequence, so $\frac{1}{nT} \sum_{t=1}^{nT} \ell(\theta, z_t) = \frac{1}{nT} \sum_{j=1}^{n} \sum_{t=1}^{T} \ell(\theta, z_{j,t})$. Expectation alignment (concatenation with resets) gives

$$\mathbb{E}_{\widetilde{\mathsf{P}}^{(nT)}} \left[ \frac{1}{nT} \sum_{t=1}^{nT} \ell(\theta, z_t) \right] = \frac{1}{n} \sum_{j=1}^{n} \mathbb{E}_{\mathsf{P}_j^{(T)}} \left[ \frac{1}{T} \sum_{t=1}^{T} \ell(\theta, z_{j,t}) \right].$$

Substitute these two equalities into the definition of $\widetilde{Y}_\theta$ to obtain $X_\theta = \widetilde{Y}_\theta$ pointwise, hence the equality of suprema. □

*Proof of Theorem 4.4.* Recall from Lemma E.23, it is sufficient to bound $\sup_\theta |\widetilde{Y}_\theta|$.

Now, to take supremum over $\widetilde{Y}_\theta$, we get

$$\sup_{\theta \in \Theta} X_\theta = \sup_{\theta \in \Theta} \frac{1}{n} \sum_{j=1}^{n} Y_{\theta, \mathsf{P}_j^{(T)}}$$

$$\leq \frac{1}{n} \sum_{j=1}^{n} \sup_{\theta \in \Theta} Y_{\theta, \mathsf{P}_j^{(T)}}.$$

To upper bound $\sup |\widetilde{Y}_\theta|$, we take a similar approach to (5, Proposition A.4).

Assuming the index set $\Theta$ is equipped with a distance metric $\rho$ and diameter $D$. We assume that for any ball $\Theta'$ of radius $r$ in $\Theta$, there exists some constant $C_1$ such that the covering number admits upper bound

$$\log N(\delta, \Theta', \rho) \leq d \log(2Ar/\delta),$$

for all $0 < \delta \leq 2r$.

Now we select $\Theta_0$ such that it is a $(D_0/2)$-covering of $\Theta$. The above assumption guarantees us that we can have a $\Theta_0$ such that $\log |\Theta_0| \leq d \log(2AD/D_0)$. By Corollary E.13, $\widetilde{Y}_\theta$ is a $2B_x^2/(1-\alpha)$-subgaussian ($\alpha = \alpha(\mathsf{P}^{(nT)})$). Then, with probability at least $1 - \delta/2$,

$$\sup_{\theta \in \Theta_0} |\widetilde{Y}_\theta| \leq C \frac{2B_x^2}{(1-\alpha)} \sqrt{d \log(2AD/D_0) + \log(2/\delta)}.$$

Note that the uniform bound for independent subgaussian random variables still applies here as for each $\theta$, we are re-sampling a new chain from a new distribution sampled from $\pi$.

Assume that $\Theta_0 = \{\theta_1, \cdots, \theta_n\}$. Now for each $j \in [m]$, we consider $\Theta_j$ is the ball centered at $\theta_j$ of radius $D_0$ in $(\Theta, \rho)$. With Theorem D.3, for each process $\{\widetilde{Y}_\theta\}_{\theta \in \Theta_j}$, then

$$\psi = \psi_2, \quad \left\|\widetilde{Y}_\theta - \widetilde{Y}_{\theta'}\right\|_\psi \leq \frac{B^1}{\sqrt{nT}} \rho(\theta, \theta'),$$

where $\ell(\theta, \boldsymbol{z}) - \ell(\theta', \boldsymbol{z})$ is a $B^1 \rho(\theta, \theta')$-subgaussian random variable.

We then get

$$P\left(\sup_{\theta, \theta' \in \Theta_j} |\widetilde{Y}_\theta - \widetilde{Y}_{\theta'}| \leq C' B^1 D_0 \left(\sqrt{\frac{d \log(2A)}{nT}} + t\right)\right) \leq 2\exp(-nTt^2), \quad \text{for all } t \geq 0.$$

If we further take $t \leq \sqrt{\log(2m/\delta)/Tn}$, then with probability at least $1 - \delta/2$, it holds that for all $j \in [m]$,

$$\sup_{\theta, \theta' \in \Theta_j} |\widetilde{Y}_\theta - \widetilde{Y}_{\theta'}| \leq C' B^1 D_0 \sqrt{\frac{2d \log(2AD/D_0) + \log(4/\delta)}{nT}}.$$

By chaining, we have

$$|\widetilde{Y}_\theta| \leq |\widetilde{Y}_{\theta_j}| + |\widetilde{Y}_\theta - \widetilde{Y}_{\theta_j}|.$$

Hence with probability at least $1 - \delta$, it holds that

$$\sup_{\theta \in \Theta} |\widetilde{Y}_\theta| \leq \sup_{\theta \in \Theta_0} |\widetilde{Y}_\theta| + \sup_j \sup_{\theta \in \Theta_j} |\widetilde{Y}_\theta - \widetilde{Y}_{\theta_j}| \leq C''\left(\frac{2B_x^2}{(1-\alpha)} + B^1 D_0\right)\sqrt{\frac{d \log(2AD/D_0) + \log(2/\delta)}{nT}}.$$

Next by taking $D_0 = D/\kappa, \kappa = 1 + B^1 D\frac{(1-\alpha)}{2B_x^2}$, we get

$$\sup_{\theta \in \Theta} |\widetilde{Y}_\theta| \leq C''\left(\frac{2B_x^2}{(1-\alpha)} + B^1 D\kappa\right)\sqrt{\frac{d \log(2A\kappa) + \log(2/\delta)}{nT}}.$$

Last, we check whether the assumptions we make above hold for our function class $\ell_\Theta$. Below, we slightly abuse our notation by using $D$ as the dimension for weight matrices in $\text{TF}_{\boldsymbol{\theta}}$. By Lemma E.10, it holds that

$$\log N(\delta, B_{||\cdot||_{\text{op}}}(r), ||\cdot||_{\text{op}}) \leq L(3MD^2 + DD' + 2)\log(1 + 2r/\delta),$$

where $B_{||\cdot||_{\text{op}}}(r)$ is a ball of radius $r$ under norm $||\cdot||_{\text{op}}$.

We check that

$$||\ell(\theta, \boldsymbol{z}) - \ell(\theta', \boldsymbol{z})|| \leq B_x(LB_H^{L-1} B_\Theta) ||\theta - \theta'||_{\text{op}},$$

where it is a direct result from Proposition E.8. By plugging all the parameters, we get

$$\sup_{\theta \in \Theta} |\widetilde{Y}_\theta| \leq C\left(\frac{B_x^2}{(1-\alpha)}\right)\sqrt{\frac{L(3MD^2 + DD' + 2)\iota + \log(2/\delta)}{nT}},$$

where $\iota = \log(2 + 2(LB_H^{L-1} B_\Theta)B\frac{1-\alpha}{B_x})$

By plugging the ERM $\widehat{\boldsymbol{\theta}}$, we get

$$L(\widehat{\boldsymbol{\theta}}) \leq \inf_\theta L(\theta) + 2\sup_\theta |\widetilde{Y}_\theta|.$$

Finally, apply Lemma E.15, we have $\alpha \leftarrow \alpha^n$.

$\square$

## E.5 ANALYSIS OF COROLLARY 4.6

**Definition E.24** (Markov Random Field (MRF) with pairwise potentials). *The random vector* $\mathcal{Z} = (\mathcal{Z}_1, \cdots, \mathcal{Z}_d)$ *over* $Z^d$ *is an MRF with pariwise potentials if there exist functions* $\psi_i : Z \to \mathbb{R}$ *and* $\varphi_{ij} : Z^2 \to \mathbb{R}$ *for* $i \neq j \in \{1, \cdots, d\}$ *such that for all* $z \in Z^d$,

$$\mathbb{P}_{z \sim \mathrm{P}^d}[\mathcal{Z} = z] = \prod_{i=1}^{d} e^{\psi(\mathcal{Z}_i)} \prod_{1 \leq i < j \leq d} e^{\varphi_{ij}(\mathcal{Z}_i, \mathcal{Z}_j)}$$

*The functions* $\psi_i$ *are called as element-wise potentials and* $\phi_{ij}$ *are pairwise potentials.*

**Definition E.25.** *Given an MRF* $\mathcal{Z}$ *with potentials* $\{\varphi_i\}$ *and* $\{\psi_{ij}\}$, *we define*

$$\beta_{i,j}(\mathcal{Z}) := \sup_{\mathcal{Z}_i, \mathcal{Z}_j \in Z} |\varphi_{ij}(\mathcal{Z}_i, \mathcal{Z}_j)|; \quad \beta(\mathcal{Z}) := \max_{1 \leq i \leq d} \sum_{j \neq i} \beta_{ij}(\mathrm{P}^d).$$

**Lemma E.26.** *Given an MRF* $z$ *with pairwise potentials, for any* $i \neq j$, $I_{j \to i}(z) \leq \beta_{j,i}(z)$. $I_{j \to i}(\mathcal{Z}) \leq I_{j,i}^{\log}(\mathcal{Z}) \leq \beta_{j,i}(\mathcal{Z})$

Lemma E.26 implies that to satisfy the condition $\alpha^{\log}(\cdot) < 1/2$, it is sufficient to show that $\beta(\cdot) < 1/2$, leading to the following condition.

$$\langle \boldsymbol{w}\boldsymbol{x}_t, \boldsymbol{x}_{t+1} \rangle < \ln \frac{1}{2} + (\sigma_\epsilon^2). \tag{E.4}$$

Observe that

$$\langle \boldsymbol{w}\boldsymbol{x}_t, \boldsymbol{x}_{t+1} \rangle \leq ||\boldsymbol{w}|| \cdot \max_t ||\boldsymbol{x}_t||$$
$$= B_w B_x$$
$$< \ln \frac{1}{2} + (\sigma_\epsilon^2) \sim 0.3.$$

### E.5.1 FURTHER ANALYSIS ON DOBRUSHIN'S CONDITION ON COMMON MODELS

Here we derive the conditions for (1) ARMA and (2) VAR (with certain stability) to satisfy the Dobrushin's condition.

**ARMA.** Consider the scalar $ARMA(p, q)$ model defined as

$$x_t = \sum_{k=1}^{p} \phi_k x_{t-k} + \varepsilon + \sum_{\ell=1}^{q} \theta_\ell \varepsilon_{t-\ell}, \quad \varepsilon_t \sim \mathcal{N}(0, \sigma_\varepsilon^2).$$

Similar to the above derivation, we assume

$$|x_t| \leq B_x, \quad \forall t.$$

Its joint log-density as Gaussian MRF is

$$p(x_{1:T} \propto \exp\left(-\frac{1}{2\sigma_\varepsilon^2} \sum_{t=1}^{T} \varepsilon(x_{1:T})^2\right) = \exp\left(-\frac{1}{2\sigma_\varepsilon^2} x_{1:T}^\top J x_{1:T}\right),$$

where $J$ is a symmetric, banded precision matrix. Decomposing the quadratic form under MRF, we have

$$\phi_{ij}(x_i, x_j) = -\frac{1}{2\sigma_\varepsilon^2} J_{ij} x_i x_j, \quad i \neq j.$$

The pairwise potentials is bounded by

$$|\phi_{ij}(x_i, x_j)| \leq \frac{1}{2\sigma_\varepsilon^2} |J_{ij}||x_i x_j| \leq \frac{B_x^2}{2\sigma_\varepsilon^2} |J_{ij}|.$$

Therefore,

$$\beta_{ij}(X) := \sup_{x_i, x_j} |\phi_{ij}(x_i, x_j)| \leq \frac{B_x^2}{2\sigma_\varepsilon^2} |J_{ij}|,$$

and

$$\beta(X) = \max_i \sum_{j \neq i} \beta_{ij}(X) \leq \frac{B_x^2}{2\sigma_\varepsilon^2} \max_i \sum_{j \neq i} |J_{ij}| = \frac{B_x^2}{2\sigma_\varepsilon^2} ||J||_{\infty,\text{off}}.$$

From Lemma E.26, we require

$$\alpha_{\log}(X) \leq \beta(X).$$

Thus a sufficient condition is

$$\frac{B_x^2}{2\sigma_\varepsilon^2} ||J||_{\infty,\text{off}} < \frac{1}{2}.$$

Further, if we use the $\text{AR}(\infty)$ representation:

$$x_t = \sum_{m=1}^{\infty} \psi_m x_{t-m} + \widetilde{\varepsilon}_t,$$

with absolutely summable $\{\psi_m\}$, $J$ satisfies the crude bound

$$||J||_{\infty,\text{off}} \leqslant \sum_{m=1}^{\infty} |\psi_m|.$$

Hence the sufficient condition is also

$$\alpha_{\log}(X) < \frac{1}{2}, \quad \text{whenever} \quad \frac{B_x^2}{\sigma_\varepsilon^2} \sum_{m=1}^{\infty} |\psi_m| \text{ is sufficiently small.}$$

**VAR($p$).** Consider the vector autoregressive model

$$X_t = \sum_{k=1}^{p} A_k X_{t-k} + \varepsilon_t, \qquad \varepsilon_t \sim \mathcal{N}(0, \Sigma_\varepsilon),$$

with $X_t \in \mathbb{R}^d$. Assume the usual stability condition: the spectral radius of the companion matrix is strictly less than 1, ensuring a unique stationary Gaussian process. Throughout this section we restrict attention to a bounded domain

$$\|X_t\|_\infty \leq B_x, \qquad \forall t.$$

The joint density of the block $X_{1:T}$ can be written as

$$p(X_{1:T}) \propto \exp\left( -\frac{1}{2} \sum_{t=1}^{T-1} \left( X_{t+1} - \sum_{k=1}^{p} A_k X_{t+1-k} \right)^\top \Sigma_\varepsilon^{-1} \left( X_{t+1} - \sum_{k=1}^{p} A_k X_{t+1-k} \right) \right).$$

Expanding the quadratic term gives

$$\left( X_{t+1} - \sum_{k=1}^{p} A_k X_{t+1-k} \right)^\top \Sigma_\varepsilon^{-1} \left( X_{t+1} - \sum_{k=1}^{p} A_k X_{t+1-k} \right)$$

$$= X_{t+1}^\top \Sigma_\varepsilon^{-1} X_{t+1} \ - \ 2 \sum_{k=1}^{p} X_{t+1}^\top \Sigma_\varepsilon^{-1} A_k X_{t+1-k} \ + \ (\text{terms not involving } X_{t+1}).$$

The cross-term between $X_{t+1}$ and $X_{t+1-k}$ is

$$X_{t+1}^\top \Sigma_\varepsilon^{-1} A_k X_{t+1-k} = \sum_{i=1}^{d} \sum_{j=1}^{d} \left( \Sigma_\varepsilon^{-1} A_k \right)_{ij} X_{t+1}^{(i)} X_{t+1-k}^{(j)}.$$

Hence, in the Gaussian Markov random field representation, the pairwise potential between node $(t+1, i)$ and node $(t+1-k, j)$ is

$$\phi_{(t+1,i),(t+1-k,j)} \left( x_{t+1}^{(i)}, x_{t+1-k}^{(j)} \right) = c_k \left( \Sigma_\varepsilon^{-1} A_k \right)_{ij} x_{t+1}^{(i)} x_{t+1-k}^{(j)},$$

where $c_k$ is an absolute constant (absorbing the factor $\frac{1}{2}$). On the bounded domain $\|X_t\|_\infty \le B_x$,

$$\left| \phi_{(t+1,i),(t+1-k,j)}\big( x_{t+1}^{(i)}, x_{t+1-k}^{(j)} \big) \right| \le C\, B_x^2 \left| \big( \Sigma_\varepsilon^{-1} A_k \big)_{ij} \right|$$

for some absolute constant $C$. Thus,

$$\beta_{(t+1,i),(t+1-k,j)}(X) \;\le\; C\, B_x^2 \left| \big( \Sigma_\varepsilon^{-1} A_k \big)_{ij} \right|.$$

For a fixed node $(t+1,i)$, summing over all its neighbors yields

$$\sum_{k=1}^{p} \sum_{j=1}^{d} \beta_{(t+1,i),(t+1-k,j)}(X) \;\le\; C\, B_x^2 \sum_{k=1}^{p} \sum_{j=1}^{d} \left| \big( \Sigma_\varepsilon^{-1} A_k \big)_{ij} \right|.$$

Taking the maximum over all $(t,i)$, we obtain

$$\beta(X) \;\le\; C\, B_x^2 \sum_{k=1}^{p} \left\| \Sigma_\varepsilon^{-1} A_k \right\|_\infty,$$

where

$$\|M\|_\infty = \max_i \sum_{j=1}^{d} |M_{ij}| \qquad \text{(row-sum norm).}$$

By Lemma E.26, the log-Dobrushin coefficient satisfies

$$\alpha_{\log}(X) \;\le\; \beta(X).$$

Therefore, a sufficient condition for $\alpha_{\log}(X) < \frac{1}{2}$ is

$$C\, B_x^2 \sum_{k=1}^{p} \left\| \Sigma_\varepsilon^{-1} A_k \right\|_\infty < \frac{1}{2}.$$

If $\Sigma_\varepsilon = \sigma_\varepsilon^2 I_d$, this simplifies to

$$\frac{B_x^2}{\sigma_\varepsilon^2} \sum_{k=1}^{p} \|A_k\|_\infty < c_0,$$

for a numerical constant $c_0$.

### E.6 ADDITIONAL DETAILS

**The History Matrix.** The matrix form of $\mathtt{A}_i(q)$ is presented below

$$\mathtt{A}_i(q) := \begin{bmatrix} x_1^i & x_2^i & \cdots & x_t^i & x_{t+1}^i & x_{t+2}^i & \cdots \\ x_T^i & x_{T-1}^i & \cdots & x_{t-1}^i & x_t^i & x_{t+1}^i & \cdots \\ x_{T-1}^i & x_{T-2}^i & \cdots & x_{t-2}^i & x_{t-1}^i & x_t^i & \cdots \\ \vdots & \vdots & \vdots & \vdots & \vdots & \vdots & \cdots \\ x_{T-q}^i & x_{T-q+1}^i & \cdots & x_{t-q}^i & x_{t-q+1}^i & x_{t-q+2}^i & \cdots \end{bmatrix} \tag{E.5}$$

# F EXPERIMENTAL DETAILS

## F.1 ENVIRONMENT

We mostly train our model on NVIDIA-H100 GPUs with 2 cores each with 128GB RAM. 2 GPUs are sufficient for all of our experiments. We use PyTorch 2.1 and our code is based on the open source published by (40). Training and evaluate takes roughly 12 hours for one run.

## F.2 MODEL ARCHITECTURE

For most of our experiments, we use MOIRAI-base model. The hyperparameters are listed in Table 2.

Table 2: Hyperparameters

| parameter | values |
|---|---|
| batch size | 64 |
| loss | MSE |
| initial learning rate | 1e-3 |
| learning rate decay | cosine annealing |
| hidden dimension | 768 |
| MLP dimension | 3072 |
| number of heads | 12 |
| training steps | $20k$ |
| max sequence length | 512 |
| optimizer | AdamW |
| beta $(\beta_1, \beta_2)$ | $(0.9, 0.98)$ |
| weight decay | 1e-1 |
| warm up steps (linear) | $10k$ |

## F.3 SYNTHETIC DATA GENERATION

We generate the AR synthetic data similar to Equation (2.1) but use normalization to stabilize the values. The parameters of synthetic data are in Table 3. Consider a sequence of data $\boldsymbol{x} \in \mathbb{R}^{d \times T} := (\boldsymbol{x}_1, \ldots, \boldsymbol{x}_T)$, where $\boldsymbol{x}_t = (x_t^1, \cdots, x_t^d) \in \mathbb{R}^d$. Assuming our target (variate of interest) is in dimension 1, we assume the $\mathtt{AR}_d(q)$ process generates $x_t^1$ as follows:

$$x_t^1 = \frac{1}{qd} \sum_{i=1}^{q} \sum_{j=1}^{d} a_i^j \cdot x_{t-i}^j + \epsilon_t, \tag{F.1}$$

where $\epsilon_t \sim N(0, 1)$, $a_i^j \sim N(0, 1) \in \mathbb{R}$. After recursively generating the time series, we remove its first 50 time steps as burnout. Each AR time series has a number of covariates between 1 to 5. For training data, we sampled 100 different time series, each with $20k$ time steps. For test data, we randomly generate one time series with time step $5k$, and evaluate our model on all time steps. We set $q, d \leq 5$ in our experiments.

**Seasonality.** We also conduct experiments on datasets with seasonality information. Specifically, we consider monthly information. After generating a multi-variate time series with $T$ time steps $\boldsymbol{x} \in \mathbb{R}^{d \times T}$, we then add the seasonality information. For each time step $t$, its seasonal information is

$$a \cdot \sin \frac{2\pi T}{f} \in \mathbb{R},$$

where $a \in \mathbb{R}$ is the amplitude, $f \in \mathbb{N}^+$ is the frequency which is 30 for monthly information. The whole seasonal information will be added to the time series.

## F.4 BASELINES

**Least Squares Regression.** Consider MOIRAI taking an input AR sequence $\boldsymbol{x} \in \mathbb{R}^{d \times T}$, to match our theoretical results (Theorem 3.6), we transform $\boldsymbol{x}$ into the following input-label pairs

$$\widetilde{\boldsymbol{x}}_1 = ((\boldsymbol{x}_1, \cdots \boldsymbol{x}_q), \boldsymbol{x}_{q+1})$$
$$\widetilde{\boldsymbol{x}}_2 = ((\boldsymbol{x}_2, \cdots \boldsymbol{x}_{q+1}), \boldsymbol{x}_{q+2}) \ldots$$

Table 3: Parameter of Synthetic Data

| parameter | values |
|---|---|
| lag size | $\{1, 2, 3, 4, 5\}$ |
| variance | $\text{unif}(0.1, 1)$ |
| length $(T)$ | $20k$ |
| number of covariates $(d)$ | $\{1, 2, 3, 4, 5\}$ |
| amplitude | $\text{unif}(0, 1.5)$ |
| frequency | $30$ |

After fitting least squares on this transformed dataset with $T - q$ samples, it predicts the $T + 1$-th time step with the following input

$$\widetilde{\boldsymbol{x}}_{\text{test}} = (\boldsymbol{x}_{T-q+1}, \cdots \boldsymbol{x}_T).$$

For least squares, we use learning rate as $0.1$, and perform full gradient descent with $50, 100$ iterations.

### F.5 Additional Experiments

**Seasonality Data.**   Here we present the experimental results on training transformers on seasonality data. The data generation is the same as described above. We use the same setup for seasonality data, where our training data comes from time series with $d \in \{1, 2, 3, 4, 5\}$, and $q = \{1, 2, 3, 4, 5\}$. The evaluation results on seasonality data is in Figure 2. We observe that transformers are capable of inferring data with seasonality. Note that transformers are capable of achieving nearly optimal performance, while least squares regression fails, indicating that transformers are capable of fitting a more complicated model than `AR` on a given time series.

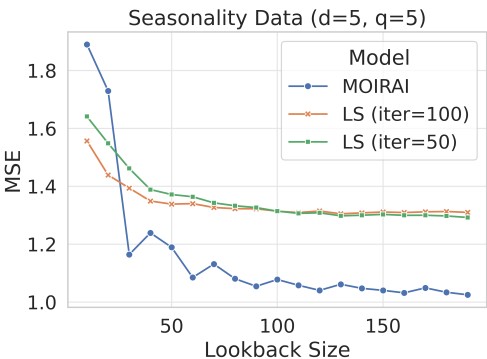

Figure 2: We observe that when least squares regression fails to obtain the optimal error rate for prediction, transformers are capable of having their MSE converge towards 1 as the lookback size increases. This indicates that these models are capable of fitting a more complex model other than linear regression on a given time series.

### F.6 Evaluation on Real-World Datasets.

Here we conduct a similar experiment to Section 5 to real-world datasets. We first pretrain a MOIRAI transformer to simulate AR regression, and then we test it on real-world datasets to see if it can handle real-world dataset in a principled way. Specifically, we use the same checkpoint of transformers in Section 5, and we pick two common datasets: ETTh1, ETTm1 and ILI to test transformers. Finally, we also include another statistical model: VARIMA as another baseline. Note that the context length here for AR models are not their lookback size, but related to their training data size as we use these AR models under the same ICL setting of MOIRAI. While we use $\text{AR}(92)$, a common lookback size for real-world datasets, we can see the performance indeed got worse.  The results are in Figure 3.

We compare the pretrained transformer with AR models with different lag size and VARIAM model. Each point in the figure is the average over 5 runs, the variance is small thus we omit it for better visualization.

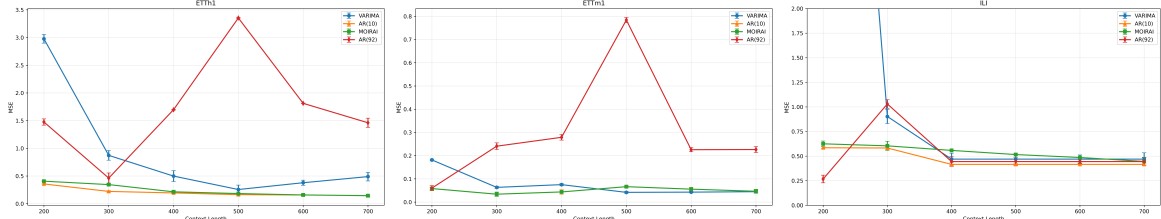

Figure 3: Comparison between pretrained transformer and AR models. While different lag size shows different performance, we are able to observe that transformers can indeed outperform all other AR models. Indicating transformers' predictive power might be more than just simulating the AR algorithm.

**Datasets.** ETTh1 and ETTm1 are datasets recording the electricity usage through out time. ETTh1 records the usage every hour, and ETTm1 records the usage every 15 minutes. The two datasets are in the same domain, but with different frequency. We test all models on the whole dataset. ILI is an Influenza Activity time series dataset of different states, is it sampled weekly.

### F.7 ABLATION STUDY ON ATTENTION BIAS

In our theory, the any-variate attention bias plays an vital role in handling arbitrary number of covariates. Here we conduct an ablation study to see whether the model performance would change without the $U$ matrix. We train MOIRAI on a mixed of different synthetic AR data. Next, we evaluate the trained MOIRAI on some other synthetic AR data with unseen $q, d$. We pretrain MOIRAI on $q \in [5, 10], d \in [5, 10]$. The results are visualized in Figure 4.

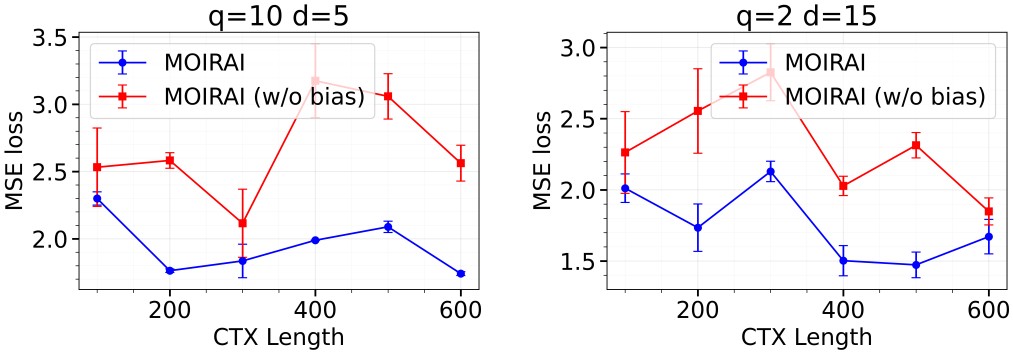

Figure 4: We observe that when using MOIRAI *with* the $U$ matrix, the performance has an obvious gain when evaluated on unseen values of $q, d$. Specifically, when $d = 15$, which is explicitly shown to MOIRAI (comparing to $q$ is a latent variable), the performance gain is even more clear.

