# OpenReview forum: "UNDERSTANDING TRANSFORMERS FOR TIME SERIES FORECASTING: A CASE STUDY ON MOIRAI"
_ICLR.cc/2026/Conference — ICLR 2026 Poster_

### Official Review · Reviewer_AzNh · 2025-10-31

**Soundness:** 3
**Presentation:** 2
**Contribution:** 2
**Rating:** 6
**Confidence:** 3

**Summary:**

The paper develops a theoretical lens on transformer models for time-series forecasting, with a focus on MOIRAI. It claims three main contributions: (i) existence proofs that transformers can implement (via in context learning -ICL-) least-squares AR regression on univariate series and—via MOIRAI’s “any-variate” mechanism—extend to multivariate AR with an arbitrary number of covariates; (ii) generalization bounds for pretraining under Dobrushin’s condition, yielding error decaying on the order of (1/$\sqrt{nT}$); and (iii) experiments (synthetic + some real-world) showing prediction error decreases with longer lookbacks, consistent with the theory. The motivation is to replace architectural heuristics with a principled understanding of why transformer time-series foundation models (particularly MOIRAI) work well and handle variable covariate counts.

**Strengths:**

In my understanding, this paper draws a new connection between transformer ICL and AR regression for time series, arguing MOIRAI’s any-variate attention and concatenation scheme enables automatic AR dimensionality selection across variable covariate sets. This is a useful theoretical complement to recent empirical TSFMs. The pretraining generalization analysis under non-IID dependence (via Dobrushin) is also timely, given TSFM pretraining on heterogeneous corpora. The positioning against prior ICL theory (largely fixed-dimensional, next-token settings) and time-series FM work is appropriate; the contribution is incremental on the ICL side but significant for time-series where clear theory is scarce.

- Bridges ICL theory to time-series AR regression; explains MOIRAI’s ability to handle arbitrary covariate counts.
- Non-IID pretraining generalization bound under Dobrushin is relevant to TSFM.
- Clear limitations section; transparent about ReLU vs softmax and AR scope.

**Weaknesses:**

The empirical section aims to validate theoretical trends rather than chase SOTA: synthetic AR data confirms error decreases with longer context and that pretrained MOIRAI adapts across (d,q); there’s a limited real-world section in the appendix. I would like to see stronger baselines beyond LSR (e.g., well-tuned ARIMA/ETS, simple RNN/Temporal-Conv) and ablation isolating any-variate bias terms’ role in practice. Also, reporting statistical variability (multiple seeds) and calibration (since pretraining uses MSE) would strengthen claims.

- Empirical findings can be extended: limited real data, modest baseline coverage, lacking variance/calibration analysis.
- The Dobrushin assumption is nontrivial; guidance on when it holds in common TS domains would help external validity.
- Several results rest on formatting assumptions and constructed sequences; practical robustness across typical TS preprocessing is less clear.

**Questions:**

- Can you characterize classes of TS (e.g., ARMA, VAR with certain stability) that satisfy Dobrushin and provide diagnostics to check it in practice?


- How sensitive are your AR-via-ICL constructions to positional encoding choices and patching (MOIRAI uses patching by default)? Can you provide ablations?


- Is it possible to add stronger empirical baselines and calibration (e.g., CRPS or PIT histograms) on real datasets to complement theory?


- Beyond AR, can your constructions cover state-space or seasonal/trend components? Even partial results would broaden scope.


- Where does any-variate bias (u₁, u₂) matter most? An ablation isolating those terms would be informative.

---

> ### Author Response · Authors · 2025-11-23
>
> Thank you for taking the time to review our submission and for providing thoughtful feedback. We appreciate your efforts and the opportunity to clarify several points raised in your review. The following is our response, and the revision includes corresponding updates (marked in BLUE). We look forward to a nice discussion with the reviewer!
>
> **W1: Empirical findings can be extended: limited...**
> > **Authors response:** In our updated manuscript, we have included (1) a new real-world dataset (ILI), (2) another statistical model baseline (VARIMA) and (3) errorbars. Note that we excluded the errorbars during submission as it is negligible.
>
> ___
>
> **W2: The Dobrushin assumption is nontrivial; guidance on when it holds in common TS domains would help external validity.**
> > **Authors response:** We provide the conditions for several TS models that satisfy the Dobrushins uniqueness condition in appendix E.5.1. However, we would like to highlight that it is indeed difficult in practice to directly test a time series Dobrushins condition from raw data. A practical way is to first fit raw data into a model of choice, and then estimate its Dobrushiness via its parameters. We have added a discussion paragraph in our section 6 to clarify this choice of assumption.
>
> ___
>
> **W3: Several results rest on formatting assumptions and constructed sequences...**
> > **Authors response:** Our formatting assumptions are indeed realistic. The only difference between our setup and MOIRAI's setup is a different positional encoding strategy, the any-variate encoding part is the same (flattening multi-variate input into univariate input). However, a simple linear transformation is capable of mapping MOIRAI’s format into our format, which can be interpreted as patch embedding or absorbed to the first layer of $W_Q, W_K, W_V$.
> We have added clarification after equation (3.4) to explain our input format.
> ___
>
> **Q1: Can you characterize classes of TS (e.g., ARMA, VAR with certain stability) that satisfy Dobrushin and provide diagnostics to check it in...**
> > **Authors response:** We provide the conditions for several TS models that satisfy the Dobrushins uniqueness condition. However, we would like to highlight that it is indeed difficult in practice to directly test a time series Dobrushins condition from raw data. A practical way is to first fit raw data into a model of choice, and then estimate its Dobrushiness via its parameters.
> For the conditions of ARMA and VAR, we provide further derivations in Appendix E.5 section.
>
> ___
>
> **Q2: How sensitive are your AR-via-ICL constructions to positional encoding choices and patching (MOIRAI uses patching by default)? Can you provide ablations?**
> > **Authors response:** We respond to this question from a theoretical perspective. For patching (MOIRAI uses patching by default), it is not sensitive in our study as an AR process remains an AR process after linear transformation (random matrix initialization).
> For the positional encoding (p), one can replace it with the standard MOIRAI time index (integer index), in which case we will require another linear transformation (Patch embedding) to map it into our desired form. Therefore, our setup is indeed general enough to handle realistic scenarios. Since patch embedding reduces sequence length by default, one can think of it represents the AR process with a higher frequency ones in the latent space. This corresponds to reduced $q_{\max}$, and further improves model's robustness to large $q$. However, if one switched from time index to sinodual positional encodings, it would require further studies to see whether it is compatible with our theory.
>
> ___

---

> > ### Author Response · Authors · 2025-11-23
> >
> > **Q4: Is it possible to add stronger empirical baselines and calibration (e.g., CRPS or PIT histograms) on real datasets to complement theory?**
> > > **Authors response:** In our experiments, we use MSE loss instead of probabilistic forecasting to match our theory. This is also highlighted in our limitation section. Therefore, we did not include those metrics in our updated manuscript.
> > ___
> > **Q5: Beyond AR, can your constructions cover state-space or seasonal/trend components? Even partial results would broaden scope.**
> > > **Authors response**: Our result covers state-space and models with seasonal components. In Theorem E.5, we show MOIRAI performs MLE via in-context learning. However, while our construction provides a solution to learning those models in-context, it is done through gradient descent. However, when comparing to other works [1] that studies transformers for MLE, we suspect our construction might not be the most parameter efficient construction.
> >
> > [1] Transformers Simulate MLE for Sequence Generation in Bayesian Networks
> > ___
> > **Q6: Where does any-variate bias (u₁, u₂) matter most? An ablation isolating those terms would be informative....**
> > > **Authors response**: We have included an ablation study on the bias terms in the appendix F.7. We observe that without $u_1, u_2$, MOIRAI did suffer from a performance degradation on unseen $d, q$.

---

### Official Review · Reviewer_Fmb5 · 2025-11-01

**Soundness:** 3
**Presentation:** 3
**Contribution:** 3
**Rating:** 6
**Confidence:** 4

**Summary:**

This work provides a clean theory for why transformer-style time series foundation models, by focusing on the MOIRAI work. They're showing a transformer can learn to perform least-squares autoregression on univariate series via in-context gradient descent. The authors then prove MOIRAI's any-variate encoding and any-variate attention let the model infer an AR predictor with an arbitrary number of covariates, explaining its strong zero-shot and few-shot behavior across domains.

**Strengths:**

I found the originality, quality, clarity and significance of the work very well. The paper gives a fresh, concrete theory for TS transformers by showing a transformer can do least-squares AR via in-context gradient descent and by explaining how MOIRAI’s any-variate attention builds per-variate lag histories to support multivariate AR, extending prior ICL views to time series with unknown lags. MOIRAI is a perfect paper in my view, and I am happy that a team is working on it. The technical core is strong, and the theory is then stress-tested on real data. The paper also defines the attention variants and the MOIRAI transformer cleanly and keeps the flow from setup easy to follow. Finally, the authors unify why MOIRAI-style FMs work in forecasting, give a generalization story for non-IID series, and show practical gains over tuned AR baselines on ETTh1/ETTm1, so the results matter both for theory and for building stronger pretrained TS models.

**Weaknesses:**

While I really enjoyed reading the methodology and approaches that the authors followed, I'd like to raise some weaknesses that came to my mind, and I'm looking forward to a nice discussion on these with the authors.

(1) From my understanding, the theory and constructions use a ReLU-based attention, and not softmax, so I'm curious to know if the authors are willing to either extend the proofs or add tests showing the key claims still hold with softmax MOIRAI.

(2) Another point is that the pretraining bounds hinge on Dobrushin's condition, so I was wondering if the authors could report a practical check for this on ETTh1/ETTm1 or at least share their thoughts on it.

(3) Additionally, I found the real-data study a bit narrow because only ETTh1/ETTm1 and comparisons only to AT models were discussed, so maybe adding another domain and strong time series foundation model baselines would make the empirical case much stronger. Do the authors have any thoughts on it? Would be good to know.

(4) Finally, the loss uses output clipping (ClipBx), and theory/results depend on qmax and dmax, so I was thinking maybe a short sensitivity study and guidance on choosing these would improve usability.

**Questions:**

I'd encourage the authors to check the Weaknesses part first, and here I have two more questions.

(1) Your theory assumes d≤dmax and q≤qmax with qmax·dmax tied to the hidden size D. How should we expect the model to behave when test-time d or q exceed these limits, and can we anticipate meaningful degradation vs sharp failure from the construction? Would be good to know the answer to it

(2) Any-variate attention uses a block matrix U with fixed block size T, so I was wondering how the framework handles irregular sampling or missing timestamps across covariates where a clean block structure does not exist and what time encoding is implied in that case?

Thanks again for your hard work.

---

> ### Author Response · Authors · 2025-11-23
>
> Thank you for taking the time to review our submission and for providing thoughtful feedback. We appreciate your efforts and the opportunity to clarify several points raised in your review. The following is our response, and the revision includes corresponding updates (marked in BLUE).  We look forward to a nice discussion with the reviewer!
>
> **W1: From my understanding, the theory and constructions use a ReLU-based...**
> > **Authors response**:  *Approximation.* In general, our approximation result is largely based on the ReLU approximation theory. There is limited literature that shows the apaproximation theory for just softmax attention (without FFN). The closest one we can find is [1], which derives the approximation results on softmax attention. However, this paper requires minor modification on the attention layer and requires an exponential number of attention heads.
>
> [1] Universal Approximation with Softmax Attention (arXiv:2504.15956)
> > *Generalization.* For generalization, we will only need to switch the Lipschitz constant of ReLU to Softmax, which are both upper bounded by 1. Therefore, in principle, our generalization bound is compatible with softmax attention.
>
> ___
>
> **W2: Another point is that the pretraining bounds hinge on Dobrushin's condition...**
> > **Authors response:** First, we wish to highlight that Dobrushins condition is a property of joint distribution, instead of the raw time series data. In general, it is difficult to test Dobrushin’s condition on raw time series data but can only test it on TS models like AR model, ARMA, etc. The main reason is we cannot estimate the influence matrix (Def. 4.2) without knowing the distributions of input data. A practical way is to first fit TS models on ETTh1/ETTm1, and then test Dobrushin’s condition on the fitted models. We have added a detailed discussion in our section 6. Note that after performing such test, we did not find ETTh1 and ETTm1 to satisfy the Dobrushins condition under different AR models.
> However, we would like to highlight that (1) AR models might not be the right choice to reflect the true joint distribution of variables, and (2) data distributions that do not satisfy Dobrushins condition might still be learnable, the use of this assumption is to simply exclude unlearnable examples in theory.
> ___
>
> **W3: Additionally, I found the real-data study a bit narrow because only ETTh1/ETTm1 and comparisons only to AT models were discussed...**
> > **Authors response**: Thank you for the suggestion. The ultimate goal for our simulation is to verify our theory instead of benchmarking different models. We have added the ILI dataset and also increased the lookback lag for all real world time series to 96, which is more realistic and closer to common setups. For stronger TSFM models such as MOIRAI-MoE and MOIRAI2, we leave it to future analysis.
> ___
> **W4: Finally, the loss uses output clipping (ClipBx), and theory/results depend on qmax and dmax, so I was...**
> > **Authors response**: 	The main limitation of qmax and dmax is the hidden dimension depends on those two variables, which relates to MOIRAI’s ICL-like capability. However, in practice, MOIRAI’s hidden size is way larger than the dimensions of most time series in the LOTSA dataset. For qmax, it indicates the max lag MOIRAI can handle, which is impacted by two factors: (a) the intrinsic lag in the time series and (b) the patch embedding size. Note that given a time series with lag $q$, patch embedding indeed shortens the sequence length of it, making it appear like a higher frequency time series in the hidden dimension. Therefore, given an input sequence with length 512, with patch size 8, a hidden dimension of 384 (MOIRAI-small) would be sufficient to solve AR up to 6 covariates. The use of output clipping is mainly for technical reasons. The goal is to obtain the subGaussian bound in our Corollary E.13. One can simply use our Theorem E.12 to obtain the same bound without the loss clipping.
> ___

---

> ### Author Response · Authors · 2025-11-23
>
> **Q1: Your theory assumes d≤dmax and q≤qmax with qmax·dmax tied to the hidden size D. How should we expect the model to behave when test-time**
> > **Authors response:**  Indeed the max number of lags and covariates depends on the hidden size $D$ If MOIRAI takes more than $d_{max}$ covariates, it will only fit AR with the first $d_{max}$ covariates. Therefore, while exceeding $d_{max}$ will not completely destroy model performance, a certain performance degradation is expected. The same concept applies to $q_{max}$ as well.
> However, we would like to highlight a counterexample that one can find an AR model that has non-zero weights on > $q_{max}$ lags, in this case, the model might not be able to generate meaningful prediction.
> ___
>
> **Q2: Any-variate attention uses a block matrix U with fixed block size T....**
> > **Authors response:** In general, we believe MOIRAI is designed for and pretrained on regularly sampled time series.  Thus it does not have a designed mechanism for this issue (not to our best knowledge). However, If the data points are missing on the grid, then our suggestion is to pad the input data so the positional relationships are fixed. Or the time index should follow the actual temporal gap. Another equivalent solution would be passing a time index with the actual temporal distance.
> However, if the irregular sampling is beyond the grid (continuous temporal distance), then our construction cannot handle this case. In our pretraining analysis, when a time series is generated by Dobrushin’s distribution (or even any mixing properties, etc), dependencies between variables (data points) are affected by their temporal distances. Therefore, from a high-level perspective, passing the temporal distances to the model is necessary to estimate the true distribution of the data.
>
> ___
>
> We thank the reviewers again for their effort and insightful feedbacks, we look forward to hear back from you!

---

### Official Review · Reviewer_TXYd · 2025-11-01

**Soundness:** 3
**Presentation:** 2
**Contribution:** 3
**Rating:** 4
**Confidence:** 2

**Summary:**

This paper provides a theoretical analysis of how transformers can perform well for time series forecasting, focusing particularly on the set of time series modeled by AR processes. First, the authors show that transformers with linear attention can approximate AR regression on univariate, as well as multivariate case (based on the any-variate attention in MOIRAI). Second, the authors use Dobrushin's condition to show a generalization bound for why pretraining on diverse datasets works. Finally, the authors ran experiments on synethetic AR data by first pretraining on generated AR data, and then compare the performances on in-distribution AR processes and out-of-distribution AR processes against least squares regression.

**Strengths:**

The paper is mostly easy to follow. It extends the ideas in Bai et al. to show that a transformer can approximate any AR model, and also shows the generalization ability of time series pretraining. It also conducts a few controlled experiments to validate the findings.

**Weaknesses:**

1. Writing. I can follow the paper, but there are quite a few typos here and there, and even in the title “Time Seires” should be “Time Series”,
- I think the following description of iTransformer might be a bit misleading: “iTransformer (Liu et al., 2023) propose to use a pooling technique to reshape arbitrary number of covariates into a unified size.”
- Line 298-299: “.” instead of “,”. For each time series, we encode it with any-variate encoding into an input matrix denoted asH ∈ RD×N , 1 We define each pretraining sample as zj := (Hj , yj ), where yj = x1T j .
- This seems to be an overstatement. I believe there are other transformer models, or TSFMs that can handle an arbitrary number of covariates. “Note that in the multi-variate case, we only focus on MOIRAI as it is the only transformer-based model that is compatible with arbitrary number of covariates.”
- Move Figure 1 closer to the experiment section.
- Line 143, w* = (w1, ..., wj ) \in R^{qd} should be  (w1, ..., wd )?
2. The proof on approximating any AR model with a transformer seems to be a straightforward application of the proof in Bai et al., as the main difference is the construction of the feature and label pairs. However, I am not very familiar with related literatures. The authors can correct me if I am wrong.

**Questions:**

1. I don’t think the setting in Appendix F6 EVALUATION ON REAL-WORLD DATASETS is appropriate. The window size seems to be too small for ETTm1 and ETTh1 to cover a whole period of time series. This might be the reason that MSE increases along with window size in Figure 3 left, which is counter-intuitive.
2. Line 80: "We impose periodic boundary conditions for the negative index, i.e., x−1 = xT." Why do we need this? It seems unconventional.
3. A recent paper "Why Do Transformers Fail to Forecast Time Series In-Context?" (https://arxiv.org/pdf/2510.09776) argues that a simple linear model has direct access to the full history of the time series, while an LSA-based model must compress this entire history into the fixed-dimensions in the Q/K/V matrices. Based on this observation, they show that for any AR(p) process, an optimally parameterized LSA model cannot achieve a lower expected MSE than the classical optimal linear predictor. Can you compare your analysis with theirs, and share your thoughts on why your findings are different?
4. Section 4 is a bit difficult to follow. Consider adding/moving more insights and intuitions to somewhere earlier in the paper.

---

> ### Author Response · Authors · 2025-11-23
>
> Thank you for taking the time to review our submission and for providing thoughtful feedback. We appreciate your efforts and the opportunity to clarify several points raised in your review. The following is our response, and the revision includes corresponding updates (marked in BLUE).
>
> **W1: Writing: I can follow the paper, but there are quite a few typos here and there...**
> > **Authors response:** We have addressed typos pointed out by the reviewer and also some other ones in our updated manuscript.
> ____
> **W1.1:I think the following description of iTransformer might be a bit misleading: ...**
> > **Authors response:** Thank you for the feedback, we agree this is misleading. We have changed it into “iTransformers treat each covariate as tokens. Compared to traditional approaches that perform attention along the temporal dimension, they perform attention along the variate dimension.
>  ____
> **W1.3:This seems to be an overstatement. I believe there ...**
> > **Authors response:**  Thank you for pointing this out, we agree this is an overstatement. We have modified it into: “Note that in the multi-variate case, we focus on MOIRAI as it is compatible with arbitrary numbers of covariates. For other transformer-based models with the same feature, we leave it to future work.”
> ____
>
> **W2: The proof on approximating any AR model with a transformer seems to be a straightforward...**
> > **Authors response:**  We thank the reviewer for the feedback. Here we first summarize the contirbution of Bai et al’s paper, and then explain the differences and new insights/results from our paper.
> Here we first summarize the contribution of Bai et al.’s paper. Their major contribution is to demonstrate there exists a transformer to perform gradient descent in-context, which provides a great starting point to provide possibile explanation on the empirical success of transformers in ICL. Their generalization results focus on pretraining on ICL tasks. However, they mainly focus on the standard transformer architecture (with ReLU activation) and IID samples. They did not cover:
> - **A.** Arbitrary number of covariates (feature dimension in their paper)
> - **B.** Any-variate encoding (different input format)
> - **C.** Any-variate attention (MOIRAI’s new architecture)
> - **D.** Non-IID learning on time series
>
> > On the other hand, we focus on MOIRAI, a specific architecture & encoding method designed to address the problem where standard transformers cannot handle an arbitrary number of covariates. Their design/approach is simple but powerful.
> We extend the results from Bai et al. to tackle the above 4 setting differences (which is realistic to MOIRAI) with our novel results. To address (B), we use our Lemma 3.2, to deomnstrate how one can apply [1]’s results to MOIRAI with minimal effort. For (A, C), we provide Lemma 3.4, which demonstrates a key insight on how any-variate attention can handle (A, C). Both Lemma 3.2 and Lemma 3.4 are novel results and not based on [1]. Finally, for non-IID time series pretraining, we introduce Dobrushin’s condition and new proving techniques (Corollary E.16, Lemma E.18) to obtain generalization on time series learning.
>
> [1] Transformers as statisticians: Provable in-context learning with in-context algorithm selection
>
> > To further clarify our contribution, we have updated our contribution in the introduction, and added a paragraph to discuss this in the related works.
> ____
> **Q1: I don’t think the setting in Appendix F6 EVALUATION ON....**
> > **Authors response:** We have updated our experiment on real-world datasets in the appendix. Specifically, we notice that either increasing the order or window size of the AR model did not increase model performance.
> Note that when we mention increasing window size, we refer to the window size under the ICL setting, which corresponds to the training data size for each least squares regression model. Thus, increasing the window size will also reduce the training data size for AR models. In our new results, we have increased the lag to 92 (common in time series analysis paper), but did not see a performance gain.
> ____
>
> **Q2: Line 80: "We impose periodic boundary conditions...**
> > **Authors response:** This is simply for convenience, as many operations in transformers involve column shifting, we use this to avoid complex notations.
> ____

---

> ### Author Response · Authors · 2025-11-23
>
> **Q3: A recent paper "Why Do Transformers Fail to Forecast Time Series In-Context?" (https://arxiv.org/pdf/2510.09776...**
> > **Authors response:**      We thank the reviewer for mentioning this paper. We first summarize the difference in setup of two papers.
> In our paper, we study the function class of MOIRAI, where [1] studies transformers with LSA with no-MLPs. Next, while in our paper, our construction also reshapes input sequence into Hankel-like matrices (sliding window features) and then performs LSR by gradient descent. On the other hand, [1] directly feeds the Hankel matrix $H$ as input and shows that the sigma-algebra generated by $LSA(H)$ is strictly coarser than generated by $H$ itself, so LSA inevitably loses information relative to OLS.
> Also, our construction is algorithmic, and [1] focuses on representation space, and our results rely on two key powers that [1] does not have (Lemma 3.2, 3.4, Prop 3.3, Thm 3.6). Further, our construction uses model depth to unroll GD steps and [1] uses model depth to compose cubic maps, which cannot simulate least square regression. Thus, the difference between our work and [1] lies in our settings and goals. Finally, we have added a short discussion on this work in the related work section.
>
> [1] Why Do Transformers Fail to Forecast Time Series In-Context?
>
> ___
>
> **Q4: Section 4 is a bit difficult to follow...**
> > **Authors response:** We have added a short introduction and summary of main results in the beginning of section 4. We added: “In this section, we investigate the generalization bound of pretraining MOIRAI on multiple non IID time series. Specifically, we assume the time series satisfy the Dobrushin's condition, which is a regularity condition commonly used for dependent data. Informally, we show that the generalization error decays roughly with the order of $O(\tfrac{1}{\sqrt{nT}})$, where $n$ is the number of time series and $T$ is the length of each time series. This provides insights and justifications on the large scale pretraining for time series transformer-based models.”
> The revised text can also be found in our revised manuscript.
>
> We thank the reviewer again for their effort and insightful feedbacks. We look forward to hear back from you!
> If you find that our response adequately addresses your concerns, we kindly ask that you consider updating your review accordingly.

---

### Author Response · Authors · 2025-11-23
**General response to reviewers**

## **General response to reviewers**

We thank the reviewers for the insightful questions and reviews. During the rebuttal, any changes or modifications made from the submitted version are highlighted in blue in our updated manuscript.

In response to the reviewers' suggestions, we have made revisions to improve the overall readability of the paper. We have added

 1. discussion on several related works (**Appendix C**)
 2. intuition and summary of section 4 in the beginning of the section
 3. suggestions to test Dobrushins condition on real world data (**Section 6 last paragraph**)
 4. discussions on sensitivity of in qmax, dmax in our construction (**Remark 3.8**)
 5. our contribution summary in introduction to highlight our contribution and what separates our work to Bai et al. (**Appendix C**)
 6. conditions for common models to satisfy Dobrushins condition (**Section E.5.1**)
 7. fixed typos

Simulation wise, we have
   1. Ablation study on any-variate attention bias  (**Appendix F.7**)
   2. Stronger baselines (RNN, AR model with realistic lag size) (**Appendix F.6**)
   3. results on ETTh1, ETTm1, ILI dataset with longer lookback window. (**Appendix F.6**)

We have conducted another round of proofreading and have fixed all typos. Several sections, tables, and figures have been updated for clarity and completeness. These revisions include additional explanations, paragraphs and sections to help readers build intuition, and visual demonstrations of the central idea and experimental results of our work.

---

### Meta-Review · Area_Chair_Sdi1 · 2026-01-07

**Summary:**

This paper provides a theoretical analysis of how transformers can perform well for time series forecasting, focusing particularly on the set of time series modeled by AR processes. Three reviewers provided mixing feedbacks on this paper. Reasons leaning for acceptance include fresh view on TS transformers, solid theoretical analysis, and good presentation. Concerns include limited comparison with SOTA baselines, narrow evaluation scope on real-world experiment, and etc. Some concerns were resolved in the rebuttal phase. I recommend acceptance of this paper, while still open for further discussion.

**Reviewer Concerns:**

Reviewer TXYd raised concerns on the writing quality, novelty and originality of the proof. The authors provided clarifications and improved the writing quality in the revised version. The author also clarified the novelty and originality of the proof. The detailed discussion should also be included in the final version.

Reviewer Fmb5 raised concerns on insufficient discussion on several findings. Addressed.

Reviewer AzNh raised concerns on readability, evaluation scope, inductive setting, and missing ablations. The authors provided clarifications and additional results to address these concerns.

**Reviewer Scores:**

Given that most concerns were resolved in the rebuttal phase, I think the reviewers' scores may be slightly increased. However, this is just an educational guess, as no explicit indication is given by the reviewers.

---

### Decision · Program_Chairs · 2026-01-26

Accept (Poster)